# Promising or Elusive? Unsupervised Object Segmentation from Real-world Single Images

**Yafei Yang     Bo Yang**

vLAR Group, The Hong Kong Polytechnic University
ya-fei.yang@connect.polyu.hk     bo.yang@polyu.edu.hk

## Abstract

In this paper, we study the problem of unsupervised object segmentation from single images. We do not introduce a new algorithm, but systematically investigate the effectiveness of existing unsupervised models on challenging real-world images. We firstly introduce four complexity factors to quantitatively measure the distributions of object- and scene-level biases in appearance and geometry for datasets with human annotations. With the aid of these factors, we empirically find that, not surprisingly, existing unsupervised models catastrophically fail to segment generic objects in real-world images, although they can easily achieve excellent performance on numerous simple synthetic datasets, due to the vast gap in objectness biases between synthetic and real images. By conducting extensive experiments on multiple groups of ablated real-world datasets, we ultimately find that the key factors underlying the colossal failure of existing unsupervised models on real-world images are the challenging distributions of object- and scene-level biases in appearance and geometry. Because of this, the inductive biases introduced in existing unsupervised models can hardly capture the diverse object distributions. Our research results suggest that future work should exploit more explicit objectness biases in the network design.

## 1 Introduction

The capability of automatically identifying individual objects from complex visual observations is a central aspect of human intelligence [53]. It serves as the key building block for higher-level cognition tasks such as planning and reasoning [28]. In last years, a plethora of models have been proposed to segment objects from single static images in an unsupervised fashion: from the early AIR [22] and MONet [8] to the recent SPACE [39], SlotAtt [40], GENESIS-V2 [20], *etc*. They jointly learn to represent and segment multiple objects from a single image, without needing any human annotations in training. This process is often called perceptual grouping/binding or object-centric learning. These methods and their variants have achieved impressive segmentation results on numerous synthetic scene datasets such as dSprites [42] and CLEVR [33]. Such advances come with great expectations that the unsupervised techniques would likely close the gap with fully-supervised methods for real-world visual understanding. However, few work has systematically investigated the true potential of the emerging unsupervised object segmentation models on complex real-world images such as COCO dataset [38]. This naturally raises an essential question:

*Is it promising or even possible to segment generic objects from real-world single images using (existing) unsupervised methods?*

**What is an object?** To answer the above question involves another fundamental question: what is an object? Exactly 100 years ago in Gestalt psychology, Wertheimer [61] first introduced a set of Gestalt principles such as proximity, similarity and continuation to heuristically define visual

36th Conference on Neural Information Processing Systems (NeurIPS 2022).

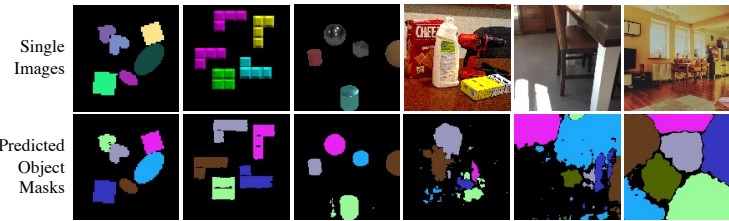

Figure 1: The failure of SlotAtt [40] on three real-world images (right-hand side), although it can perfectly segment simple objects on three synthetic images (left-hand side).

data as objects. However, these factors are highly subjective, whilst the real-world generic objects are far more complex with extremely diverse appearances and shapes. Therefore, it is practically impossible to quantitatively define what is an object, *i.e.*, the objectness, from visual inputs (*e.g.*, a set of image pixels). Nevertheless, to thoroughly understand whether unsupervised methods can truly learn objectness akin to the psychological process of humans, it is vital to investigate the underlying factors that potentially facilitate or otherwise hinder the ability of unsupervised models. In this regard, by drawing on Gestalt principles, we instead define a series of new factors to quantitatively measure the complexity of objects and scenes in Section 2. By taking into account both the appearance and geometry of objects and scenes, our complexity factors explicitly assess the difficulty of segmenting a specific object. For example, a chair with colorful textures tend to have higher complexity than a single-color ball for unsupervised methods. With the aid of these factors, we extensively study whether and how existing unsupervised models can discover objects in Section 4.

**What is the problem of unsupervised object segmentation from single images?** A large number of models [63] aim to tackle the problem of unsupervised object segmentation from single images. They share several key problem settings: 1) all training images do not have any human annotations; 2) every single image has multiple objects; 3) each image is treated as a static data point without any dynamic or temporal information; 4) all models are trained from scratch without requiring any pretrained networks on additional datasets. Ultimately, the goal of these models is to segment all individual objects as accurate as the ground truth human annotations. In this paper, we regard these settings as the basic and necessary part of unsupervised object segmentation from single images, and empirically evaluate how successfully the existing models can exhibit on real-world images.

**Contributions and findings.** This paper addresses the essential question regarding the potential of unsupervised segmentation of generic objects from real-world single images. Our contributions are:

- We firstly introduce 4 complexity factors to quantitatively measure the difficulty of objects and scenes. These factors are key to investigate the true potential of existing unsupervised models.
- We extensively evaluate current unsupervised approaches in a large-scale experimental study. We implement 4 representative methods and train more than 130 models on 6 curated datasets from scratch. The datasets, code and pretrained models are available at https://github.com/vLAR-group/UnsupObjSeg
- We analyze our experimental results and find that: 1) existing unsupervised object segmentation models cannot discover generic objects from single real-world images, although they can achieve outstanding performance on synthetic datasets, as qualitatively illustrated in Figure 1; 2) the challenging distributions of both object- and scene-level biases in appearance and geometry from real-world images are the key factors incurring the failure of existing models; 3) the inductive biases introduced in existing unsupervised models are fundamentally not matched with the objectness biases exhibited in real-world images, and therefore fail to discover the real objectness.

**Related Work.** Recently, ClevrTex [35] and the concurrent work [43] also study unsupervised object segmentation on single images. Through evaluation on (complex) synthetic datasets only, both works focus on benchmarking the effectiveness of particular network designs of baselines. By comparison, our paper aims to explore what and how the objectness distribution gaps between synthetic and real-world images incur the failure of existing models. The recent work [60] which investigates video object discovery is orthogonal to our work as the motion signals do not exist in single images.

**Scope of this research.** This paper does not investigate unsupervised object discovery on saliency maps [59], static multi-views or dynamic videos [63]. Recent methods [10; 30] requiring pretrained models on monolithic object images such as ImageNet [47] are not evaluated as well.

# 2 Complexity Factors

As illustrated in the top row of Figure 2, an individual object, represented by a set of color pixels painted within a mask, can vary significantly given different types of appearance and geometric shape. A specific scene, represented by a set of objects placed within an image, can also differ vastly given different types of relative appearance and geometric layout between objects, as illustrated in the bottom row. Unarguably, such variation and complexity of appearance and geometry in both object level and scene level directly affects human's ability to precisely separate all objects. Naturally, the performance of unsupervised segmentation models are also expected to be influenced by the variation. In this regard, we carefully define the following two groups of factors to quantitatively describe the complexity of different datasets.

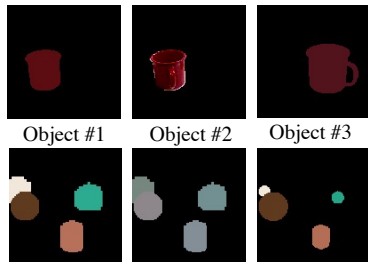

Figure 2: Complexity in appearance and geometry for objects and scenes.

## 2.1 Object-level Complexity Factors

As to a specific object, all its information can be described by appearance and geometry. Therefore we define the below two factors to measure the complexity of appearance and geometry respectively. Notably, both factors are nicely invariant to the object scale.

- **Object Color Gradient:** This factor aims to calculate how frequently the appearance changes within the object mask. In particular, given the RGB image and mask of an object, we firstly convert RGB into grayscale and then apply Sobel filter [51] to compute the gradients horizontally and vertically for each pixel within the mask. The final gradient value is obtained by averaging out all object pixels. Note that, the object boundary pixels are removed to avoid the interference of background. Numerically, the higher this factor is, the more complex texture and/or lighting effect the object has, and therefore it is likely harder to segment.
- **Object Shape Concavity:** This factor is designed to evaluate how irregular the object boundary is. Particularly, given an object (binary) mask, denoted as $M_{obj} \in \mathbb{R}^{H \times W}$, we firstly find the smallest convex polygon mask ($M_{cvx} \in \mathbb{R}^{H \times W}$) that surrounds the object mask using an existing algorithm [19], and then the object shape concavity value is computed as: $1 - \sum M_{obj} / \sum M_{cvx}$. Clearly, the higher this factor is, the more irregular object shape is, and segmentation is more tricky.

## 2.2 Scene-level Complexity Factors

As to a specific image, in addition to the object-level complexity, the spatial and appearance relationships between all objects can also incur extra difficulty for segmentation. We define the following two factors to quantify the complexity of relative appearance and geometry between objects in an image.

- **Inter-object Color Similarity:** This factor intends to assess the appearance similarity between all objects in the same image. Specifically, we firstly calculate the average color for each object, and then compute the pair-wise Euclidean distances of object colors, obtaining a $K \times K$ matrix where $K$ represents the object number. The *average color distance* is calculated by averaging the matrix excluding diagonal entries, and the final inter-object color similarity is computed as: $1 - average$ $color\ distance / (255 \times \sqrt{3})$. Intuitively, the higher this factor is, the more similar all objects appear to be, the less distinctive each object is, and it is harder to separate each object.
- **Inter-object Shape Variation:** This factor aims to measure the relative geometry diversity between all objects in the image. We firstly calculate the diagonal length of bounding box for each object, and then compute the pair-wise absolute differences for all object diagonal lengths, obtaining a $K \times K$ matrix. The final inter-object shape variation is the average of the matrix excluding diagonal entries. The higher this factor, the objects within an image have more diverse and imbalanced sizes, and therefore segmenting both gigantic and tiny objects is likely more challenging.

By capturing the appearance and geometry in both object and scene levels, the four factors are designed to quantify the complexity of objects and images. For illustration, Figure 3 shows sample images for the four factors at different values. The higher the values, the more complex the objects

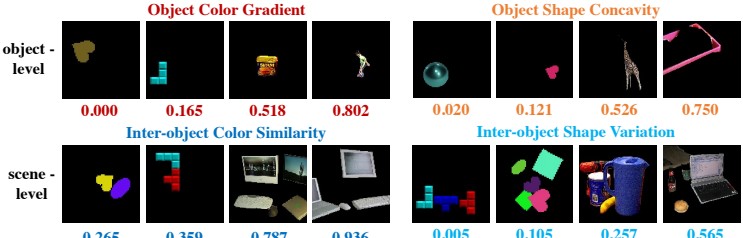

Figure 3: Sample objects and scenes for the four factors at different complexity values. All complexity values are normalized to the range of $[0, 1]$.

and scenes. In fact, these factors are carefully selected from more than 10 candidates because they are empirically more suitable to differentiate the gaps between synthetic and real-world images, and they eventually serve as key indicators to diagnose existing unsupervised models in Section 4. Calculation details of the four factors and other candidates are in appendix.

## 3 Experimental Design

### 3.1 Considered Methods

A range of works have explored unsupervised object segmentation in recent years. They are typically formulated as (variational) autoencoders (AE/VAE) [36] or generative adversarial networks (GAN) [24]. GAN based models [13; 3; 7; 56; 5; 58; 1] are usually limited to identifying a single foreground object and can hardly discover multiple objects due to the training instabilities, therefore not considered in this paper. As shown in Table 1, the majority of existing models are based on AE/VAE and can be generally divided into two groups according to the object representation:

- **Factor-based models**: Each object is represented by explicit factors such as size, position, appearance, *etc*., and the whole image is a spatial organization of multiple objects. Basically, such representation explicitly enforces objects to be bounded within particular regions.
- **Layer-based models**: Each object is represented by an image layer, *i.e.*, a binary mask, and the whole image is a spatial mixture of multiple object layers. Intuitively, this representation does not have strict spatial constrains, and instead is more flexible to cluster similar pixels as objects.

In order to decompose the input images into objects, these approaches introduce different types of network architecture, loss functions, and regularization terms as inductive biases. These biases broadly include: 1) variational encoding which encourages the disentanglement of latent variables; 2) iterative inference which likely ends up with better scene representations over occlusions; 3) object relationship regularization such as depth estimation and autoregressive prior which aims at capturing the dependency of multiple objects; and many other biases. With different combinations of these biases, many methods have shown outstanding performance in synthetic datasets. Among them, we select 4 representative models for our investigation: 1) AIR [22], 2) MONet [8], 3) IODINE [25], and 4) SlotAtt [40]. We also add the fully-supervised Mask R-CNN [29] as an additional baseline for comprehensive comparison. Implementation details are provided in appendix.

### 3.2 Considered Datasets

We consider two groups of datasets for extensive benchmarking and analysis: 1) three commonly-used synthetic datasets: dSprites [42], Tetris [34] and CLEVR [33], 2) three real-world datasets: YCB [9], ScanNet [17], and COCO [38], representing the small-scale, indoor- and outdoor-level real scenes respectively. Naturally, objects and scenes in different datasets tend to have very different types of biases. For example, the objects in dSprites tend to have the single-color bias, while COCO does not. Generally, the object-level biases can be divided as: 1) appearance biases including different textures and lighting effects, and 2) geometry biases including the object shape and occlusions. Similarly, the scene-level biases include: 1) appearance biases such as the color similarity between all objects, and 2) geometry biases such as the diversity of all object shapes. In fact, our complexity factors introduced in Section 2 are designed to well capture these biases. Table 2 qualitatively summarizes the biases of selected datasets. We may hypothesize that the large gaps of biases between synthetic and real-world datasets would have a huge impact on the effectiveness of existing models.

Table 1: Existing unsupervised models for object segmentation on single images. Each model includes different inductive biases, such as variational autoencoding (VAE), iterative inference (Iter), object relationship regularization (Rel), *etc*.

| Factor-based Models | | Inductive Biases | | | Layer-based Models | | Inductive Biases | | |
|---|---|---|---|---|---|---|---|---|---|
| | | VAE | Iter | Rel | | | VAE | Iter | Rel |
| CST-VAE [31] | ICLRW'16 | ✓ | | | Tagger [26] | NIPS'16 | | ✓ | |
| AIR [22] | NIPS'16 | ✓ | | | RC [26] | ICLRW'16 | | ✓ | |
| SPAIR [16] | AAAI'19 | ✓ | | | NEM [27] | NIPS'17 | | ✓ | |
| SuPAIR [54] | ICML'19 | ✓ | | | MONet [8] | arXiv'19 | ✓ | | |
| GMIO [64] | ICML'19 | ✓ | ✓ | | IODINE [25] | ICML'19 | ✓ | ✓ | |
| ASR [62] | NeurIPS'19 | ✓ | | | ECON [57] | ICLRW'20 | ✓ | | ✓ |
| SPACE [39] | ICLR'20 | ✓ | | | GENESIS [21] | ICLR'20 | ✓ | | ✓ |
| GNM [32] | NeurIPS'20 | ✓ | | ✓ | SlotAtt [40] | NeurIPS'20 | | ✓ | |
| SPLIT [11] | arXiv'20 | ✓ | | | GENESIS-V2 [20] | NeurIPS'21 | ✓ | | ✓ |
| OCIC [2] | arXiv'20 | ✓ | | ✓ | R-MONet [50] | arXiv'21 | ✓ | | ✓ |
| GSGN [18] | ICLR'21 | ✓ | | ✓ | CAE [41] | arXiv'22 | | | ✓ |

Table 2: The object- and scene-level biases in appearance and geometry of the considered datasets.

| | | Synthetic Datasets | | | Real-world Datasets | | |
|---|---|---|---|---|---|---|---|
| | | dSprites [42] | Tetris [34] | CLEVR [33] | YCB [9] | ScanNet [17] | COCO [38] |
| | | *Object-level Biases* | | | | | |
| Appearance | Texture: | simple | simple | simple | diverse | simple | diverse |
| | Lighting: | no | no | synthetic | real | real | real |
| Geometry | Shape: | simple | simple | simple | simple | diverse | diverse |
| | Occlusion: | minor | no | minor | severe | severe | severe |
| | | *Scene-level Biases* | | | | | |
| Appearance | Similarity: | low | low | high | high | high | high |
| Geometry | Diversity: | low | low | low | high | high | high |

To guarantee the fairness and consistency of all experiments, we carefully prepare all six datasets using the following same protocols. Preparation details for each dataset are provided in appendix.

- All images are rerendered or cropped with the same resolution of $128 \times 128$.
- Each image has about 2 to 6 solid objects with a blank background.
- Each dataset has about 10000 images for training, 2000 images for testing.

### 3.3   Considered Metrics

Having the six representative datasets and four existing unsupervised methods at hand, we choose the following metrics to evaluate the object segmentation performance: 1) AP score which is widely used for object detection and segmentation [23], 2) PQ score which is used to measure non-overlap panoptic segmentation [37], and 3) Precision and Recall scores. A predicted mask is considered correct if its IoU against a ground truth mask is above 0.5. All objects are treated as a single class. The blank background is not taken into account for fair comparison. To compute AP, we simply treat the mean value of the soft object mask as the object confidence score. Note that, the alternative metrics ARI [45] and segmentation covering (SC) [4] are not considered as they can be easily saturated.

## 4   Key Experimental Results

### 4.1   Can current unsupervised models succeed on real-world datasets?

First of all, we evaluate all baselines on our six datasets separately. In particular, we train each model from scratch on each dataset separately. For fair evaluation, we carefully tune the hyperparameters of each model on every dataset and fully optimize the networks until convergence. Figure 4 compares the quantitative results. It can be seen that all methods demonstrate satisfactory segmentation results on synthetic datasets, especially the recent strong baselines IODINE and SlotAtt. However, not surprisingly, all unsupervised methods fail catastrophically on the three real-world datasets.

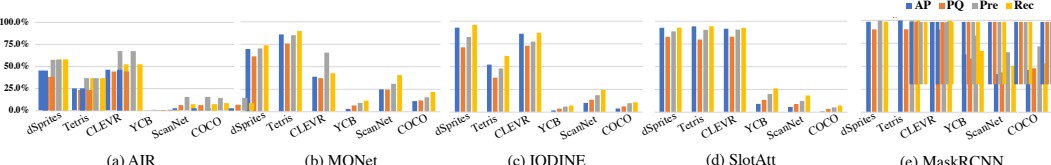

Figure 4: Quantitative results of object segmentation from the five methods on six datasets.

**Preliminary Diagnosis:** In order to diagnose the colossal failure, we hypothesize that it is because of the huge gaps in objectness biases between two types of datasets. In this regard, we quantitatively compute the distributions of our four complexity factors on the six datasets. In particular, the two object-level factors, *i.e.*, Object Color Gradient and Object Shape Concavity, are computed for each object of the six training splits. The two scene-level factors, *i.e.*, Inter-object Color Similarity and Inter-object Shape Variation, are computed for each image of the six training splits.

From the distributions shown in the top row of Figure 5, we can see that, 1) for the two object-level factors (Subfigs 1-a and 1-c), the three synthetic datasets tend to have extremely lower scores than the real-world datasets, which means that the synthetic objects are more likely have uniform colors and convex shapes; 2) for the two scene-level factors (Subfigs 1-b and 1-d), the images in synthetic datasets tend to include less similar objects in terms of color, which means that multiple objects in real-world scenes are less distinctive in appearance. In addition, the multiple objects in synthetic scenes tend to have similar sizes, whereas real-world scenes usually have diverse object scales in single images. To validate whether these distribution biases are the true reasons incurring the failure, we conduct extensive ablative experiments in Sections 4.2, 4.3 and 4.4.

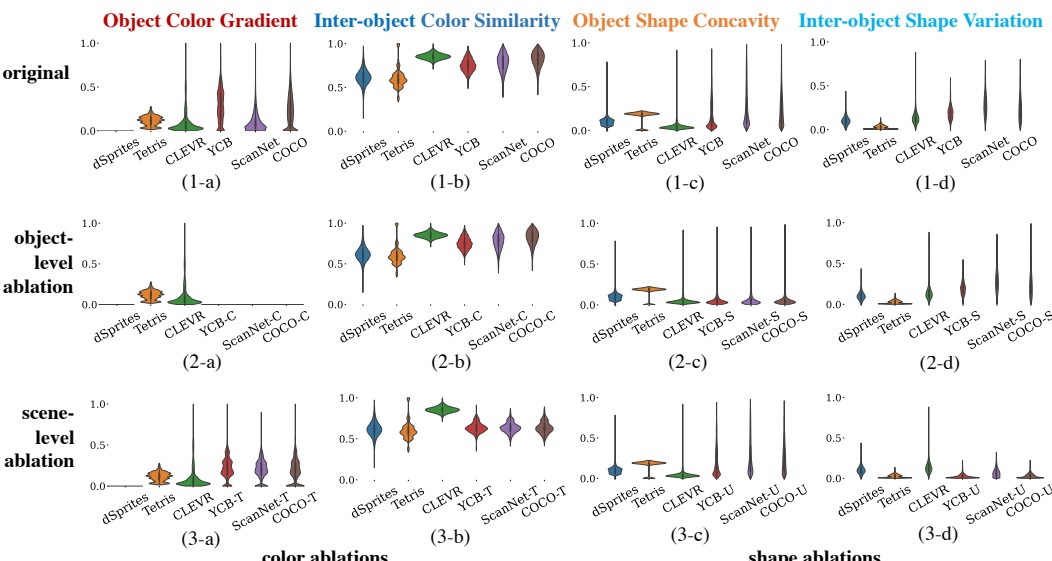

Figure 5: Distributions of the four complexity factors. The *top row* shows the distributions of original synthetic/real-world datasets in Sec 4.1. The *2nd row* shows the distributions of object-level ablated datasets in Sec 4.2. The *3rd row* shows distributions of scene-level ablated datasets in Sec 4.3.

## 4.2 How do object-level factors affect current models?

In this section, we aim to verify to what extent the distributions of object-level factors affect the segmentation performance. In particular, we conduct the following three ablative experiments.

- *Ablation of Object Color Gradient*: For each object of the three real-world datasets, we only replace all pixel colors by its average color $rgb$ value within each object mask, without touching the object shapes. In this way, the color gradients of each object are totally erased, thus removing the potential impact of Object Color Gradient. The three ablated datasets are: YCB-C / ScanNet-C / COCO-C.
- *Ablation of Object Shape Concavity*: For each object in the real-world datasets, we find the smallest convex hull [19] for its object mask and then fill the empty pixels by shifting the original object

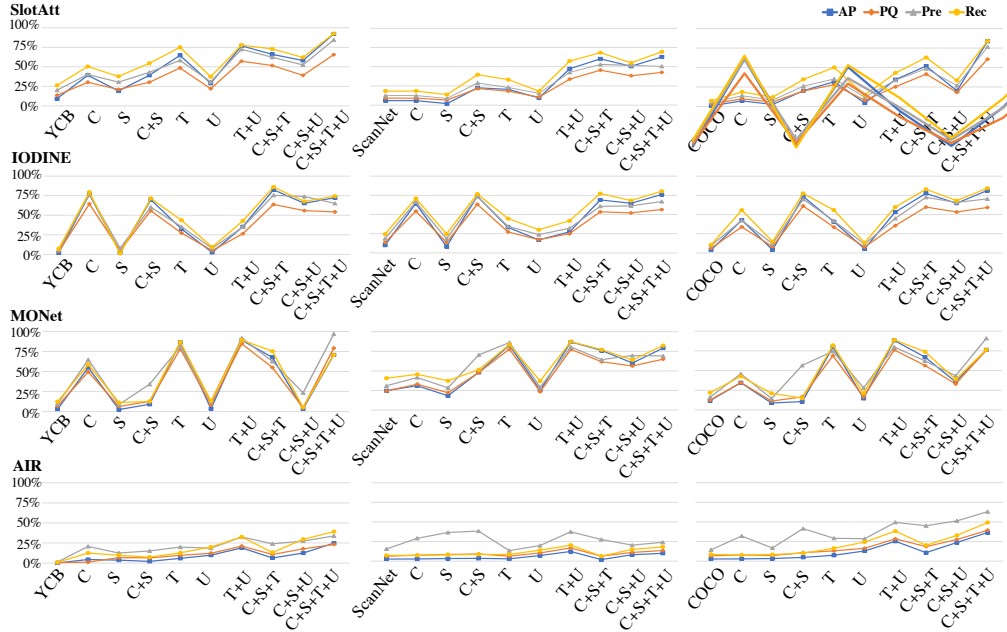

Figure 7: Quantitative results of baselines on the three real-world datasets and their variants. The letters C/S/C+S represent the three ablated datasets in Sec 4.2; T/U/T+U represent the three ablated datasets in Sec 4.3; C+S+T/C+S+U/C+S+T+U represent the three ablated datasets in Sec 4.4.

pixels. Basically, this ablation aims to only reduce the irregularity of object **s**hapes, yet retaining the distributions of color gradients. The ablated datasets are: YCB-S / ScanNet-S / COCO-S.

- *Ablation of both Object Color Gradient and Shape Concavity*: We simply combine the above two ablation for each real-world object, getting datasets: YCB-C+S / ScanNet-C+S / COCO-C+S.

For illustration, the 2nd row of Figure 6 shows example images of three ablated datasets: YCB-C / YCB-S / YCB-C+S.

In the 2nd row of Figure 5 (Subfigs 2-a/2-b), we calculate new distributions of both Object Color Gradient and Inter-object Color Similarity on the datasets YCB-C / ScanNet-C / COCO-C. We can see that the object-level gradients become all zeros in Subfig 2-a, even simpler than three synthetic datasets. Yet, the distributions of Inter-object Color Similarity are almost the same as original YCB / ScanNet / COCO, *i.e.*, Subfig 2-b is similar to 1-b.

Similarly, the 2nd row of of Figure 5 (Subfigs 2-c/2-d) shows that the distributions of Object Shape Concavity of ablated datasets now become similar to the synthetic datasets, while the distributions of Inter-object Shape Variation keep the same, *i.e.*, Subfig 2-d is similar to Subfig 1-d. Note that, for the ablation of both Object Color Gradient and Shape Concavity, the distributions will be the same as shown in the 2nd row of Figure 5. Having the three groups of object-level ablated real-world datasets, we then evaluate segmentation performance of the baselines separately.

**Brief Analysis:** As shown in Figures 7 & 9, we can see that: 1) Once the pixels of real-world objects are replaced by its mean color, *i.e.*, without any color gradients, the object segmentation performance has been significantly improved for almost all methods. 2) Reducing the irregularity of real-world objects can also improve the object segmentation, although not significantly. 3) Overall, these results show that existing methods are more likely to learn the objectness represented by uniform colors and/or regular objects. However, comparing with Figure 4, the results of

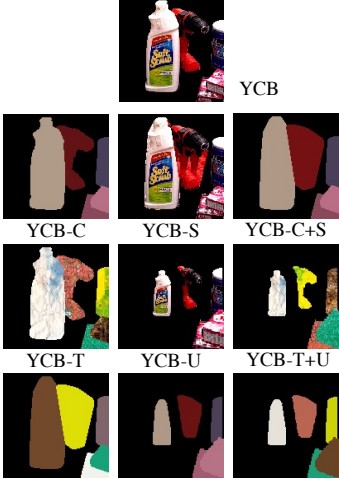

Figure 6: Sample images with different ablations. The *top row* shows an original image from YCB dataset. The *2nd row* shows examples from three ablated datasets in Sec 4.2. The *3rd row* presents examples from three ablated datasets in Sec 4.3. The last row shows examples from three ablated datasets in Sec 4.4

current ablated datasets in Figure 7 still lag behind the synthetic datasets. This means that there must be some other factors that also potentially affect the object segmentation of existing models. More results are in appendix.

### 4.3 How do scene-level factors affect current models?

In this section, we turn to investigate to what extent the distributions of scene-level factors affect the segmentation performance. Particularly, we conduct the following three ablative experiments.

- *Ablation of Inter-object Color Similarity*: In each image of the three real-world datasets, we replace all object textures by a set of new distinctive textures from the existing DTD database [15], as shown in Figure 8. In this way, the multiple objects look more distinctive in appearance, while the per-object texture gradients are roughly preserved. The ablated datasets are denoted: YCB-T / ScanNet-T / COCO-T.
- *Ablation of Inter-object Shape Variation*: In each image of the real-world datasets, we normalize the scales of multiple objects by shrinking or expanding the diagonal length of their bounding boxes, such that the new object sizes tend to be uniform. For each object, its shape and texture are linearly scaled up or down. Basically, this aims to remove the diversity of object sizes within single images. The ablated datasets are denoted as: YCB-U / ScanNet-U / COCO-U.
- *Ablation of both Inter-object Color Similarity and Shape Variation*: We simply combine the above two ablation strategies for each real-world image. Ablation details are in appendix. The ablated datasets are denoted as: YCB-T+U / ScanNet-T+U / COCO-T+U.

For illustration, the 3rd row of Figure 6 shows example images of three ablated datasets: YCB-T / YCB-U / YCB-T+U.

In the 3rd row of Figure 5 (Subfigs 3-a/3-b), we calculate new distributions of both Object Color Gradient and Inter-object Color Similarity on the ablated datasets YCB-T / ScanNet-T / COCO-T. We can see that the distributions of scene-level color similarity become more similar to synthetic datasets in Subfig 3-b, while the distributions of object color gradients are still similar as original real-world datasets YCB / ScanNet / COCO, *i.e.*, Subfig 3-a is similar to 1-a.

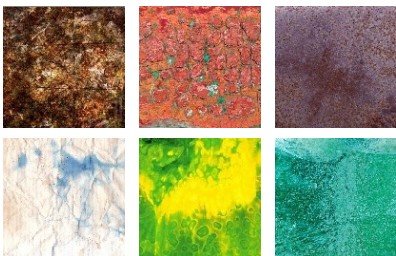

Figure 8: Six selected texture images from DTD [15].

Similarly, the 3rd row of Figure 5 shows that the distributions of Inter-object Shape Variation of ablated datasets now become similar to the synthetic datasets in Subfig 3-d, whereas the distributions of Object Shape Concavity are still the same as original real-world datasets YCB / ScanNet / COCO, *i.e.*, Subfig 3-c is similar to Subfig 1-c. Note that, for the ablation of both Inter-object Color Similarity and Shape Variation, the distributions will be the same as the 3rd row of Figure 5. Having these three groups of scene-level ablated real-world datasets, we then separately evaluate each baseline.

**Brief Analysis:** As shown in Figures 7 & 9, we can see that: 1) Once the textures of real-world images are replaced by more distinctive textures, *i.e.*, with lower similarity between object appearance, the segmentation performance has been surprisingly boosted remarkably for almost all methods. 2) Normalizing object sizes over images can also reasonably improve the segmentation performance. 3) Overall, these results clearly show that existing unsupervised models significantly favor objectness with distinctive appearance in single images. However, compared with Figure 4, the results on current scene-level ablated datasets are still inferior to synthetic datasets, meaning that the scene-level factors alone are not enough to explain the performance gap. More results are in appendix.

### 4.4 How do object- and scene-level factors jointly affect current models?

In this section, we aim to study how the object- and scene-level factors jointly affect the segmentation performance. In particular, we conduct the following three ablative experiments.

- *Ablation of Object Color Gradient, Object Shape Concavity and Inter-object Color Similarity*: In each image of the three real-world datasets, we replace the object color by averaging all pixels of the distinctive texture, and also replace the object shape with a simple convex hull. The ablated datasets are denoted: YCB-C+S+T / ScanNet-C+S+T / COCO-C+S+T.

- *Ablation of Object Color Gradient, Object Shape Concavity and Inter-object Shape Variation*: In each image of the three real-world datasets, we replace the object color by averaging its own texture, and modify the object shape as convex hull following by size normalization. The ablated datasets are denoted as: YCB-C+S+U / ScanNet-C+S+U / COCO-C+S+U.
- *Ablation of all four factors*: We aggressively combine all four ablation strategies and we get datasets: YCB-C+S+T+U / ScanNet-C+S+T+U / COCO-C+S+T+U.

For illustration, the 4th row of Figure 6 shows example images of three ablated datasets: YCB-C+S+T / YCB-C+S+U / YCB-C+S+T+U.

Since these ablations are conducted independently, the new distributions of four complexity factors on current jointly ablated datasets are exactly the same as the second and third rows of Figure 5.

**Brief Analysis:** As shown in Figure 7, we can see that: 1) Combining the two object-level factors and either of scene-level factors to ablate the real-world datasets, the segmentation performance can be improved as we expect, especially for IODINE and SlotAtt. 2) If the challenging real-world objects and images are ablated in both object- and scene-level, the segmentation performance of all unsupervised models achieves the same level with three synthetic datasets as shown in Figure 4. 3) Overall, these three groups of experiments demonstrate that the colossal failure of unsupervised models on real-world images involves both object- and scene-level dataset biases. More experiments and results are in appendix.

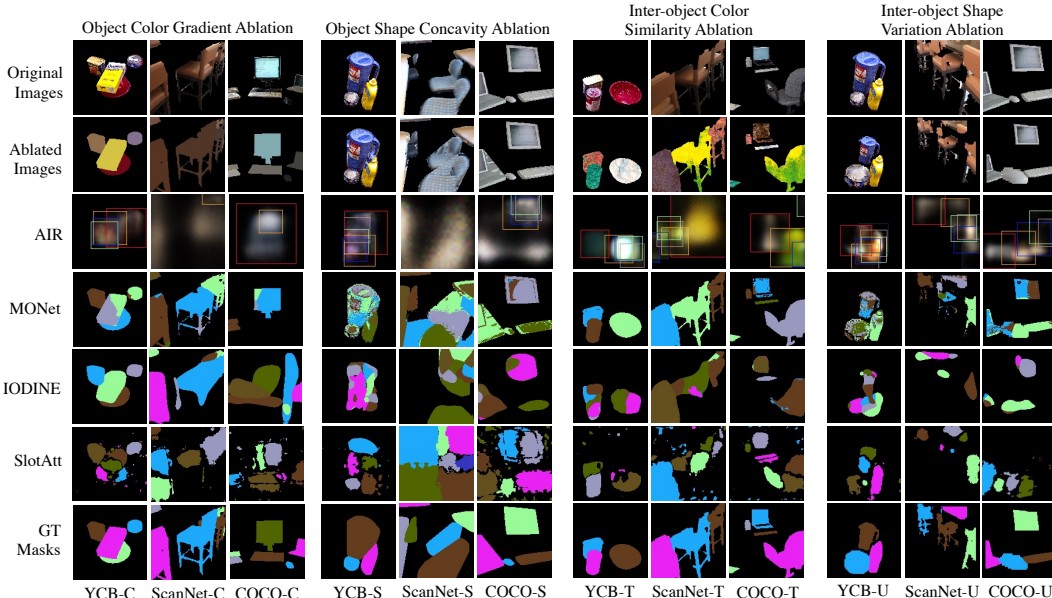

Figure 9: Qualitative results of four representative methods on multiple ablation datasets in Sec 4.2 and Sec 4.3.

## 4.5 Why do current unsupervised models fail on real-world datasets?

As demonstrated in Sections 4.2/4.3/4.4, once the complexity factors are removed from the challenging real-world datasets, existing unsupervised models can perform as excellent as on the synthetic datasets, as qualitatively illustrated in Figure 10. From this, we can safely conclude that the inductive biases designed in existing unsupervised models are far from able to match with and fully capture the true and complex objectness biases exhibited in real-world images. Nevertheless, from Figure 7, each baseline tends to favor different objectness biases. In particular,

- AIR [22]: As a factor based model, AIR has a strong spatial-locality bias. Despite its poor segmentation performance across all datasets, there is a notable improvement when inter-object shape variation is ablated from real-world datasets (U / T+U / C+S+U / C+S+T+U). More convincingly, even when all other three factors are ablated (C+S+T), it can be hardly improved. These observations show that object shape variation is a significant factor for AIR.

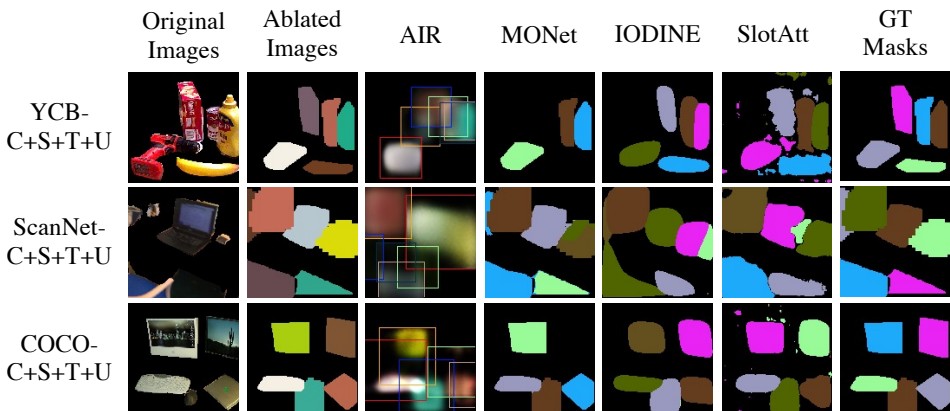

Figure 10: Qualitative results on fully ablated real-world datasets (YCB-C+S+T+U / ScanNet-C+S+T+U / COCO-C+S+T+U).

- MONet [8]: MONet is more sensitive to color-related factors than shape-related factors. The ablations of object color gradient and inter-object color similarity significantly improve its performance, while ablations of object shape concavity and inter-object shape variation make little differences. For the two color-related factors, the scene-level one is more important than the object-level factor. From this, we can see that MONet has a strong dependency on color. Similar colors tend to be grouped together while different colors are separated apart. Furthermore, the ablation on object color gradient alleviates over-segmentation whereas the ablation on inter-object color similarity alleviates under-segmentation. We conjecture that under-segmentation can be a more severe issue for MONet on real-world datasets, leading to a larger sensitivity on the scene-level color factor.

- IODINE [25]: IODINE also has a heavy dependency on both object- and scene-level color-related factors. However, different from MONet, the ablation on object color gradient brings better performance than inter-object color similarity. We speculate it is because the regularization on shape latent alleviates under-segmentation by biasing towards more regular shapes. In this way, over-segmentation is the key issue, making the object color gradient a dominant factor.

- SlotAtt [40]: The ablations on all four factors increase the performance of SlotAtt at different levels, among which object- and scene-level color-related factors are more significant. We conjecture that it is because the feature embeddings used by Slot Attention module are learnt from both pixel colors and coordinates, which contributes to its sensitivity to both shape and color factors.

## 5   Conclusions

We systematically show that existing unsupervised methods are practically impossible to segment generic objects from single real-world images, and investigate the underlying factors that incur the catastrophic failure. With the aid of our carefully designed four object- and scene-level complexity factors, we conduct extensive experiments on multiple groups of ablated real-world objects and images, and safely conclude that the distributions of both object- and scene-level biases in appearance and geometry of real-world datasets are particularly diverse and indiscriminative, such that current unsupervised models cannot segment real objects. Based on this finding, we suggest two main directions for future study: 1) To exploit more discriminative objectness biases such as object motions which expressively describe the ownership of visual pixels as recently explored in [55; 12; 6] for 2D images and in [52] for 3D point clouds. 2) To leverage pretrained features from single-object-dominant datasets which explicitly regard each image as an object as recently studied in [10; 30], although such settings are no longer purely unsupervised.

**Broader Impact:** One limitation of this work is the lack of study about foreground-background segmentation, but our work can still have a positive impact for researchers to design more practically useful models. A potential negative impact could be using our findings to attack existing models.

**Acknowledgements:** This work was partially supported by Shenzhen Science and Technology Innovation Commission (JCYJ20210324120603011).

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
