# A  Appendix

## A.1  Details of the Four Complexity Factors

We have introduced four complexity factors in both object- and scene-level appearance and geometry. Details of these factors are as follows.

- **Object Color Gradient:** As shown in Figure 11, given an RGB image of an object, we first convert it to grayscale by applying $Y = 0.299R + 0.587G + 0.114B$. Then Sobel filter [51] with kernel size $3 \times 3$ is applied horizontally and vertically to compute the image gradient for each pixel. Since this factor should only be related to the appearance inside an object regardless of the background, we remove the gradients at the object boundary. The object boundary is computed from its mask, following the practice of [14]. The factor score is calculated as the average gradient of all object pixels that are not on the boundary. This value ranges between 0 and 255, which is then divided by 255 so as to be normalized to the range of $[0, 1]$.

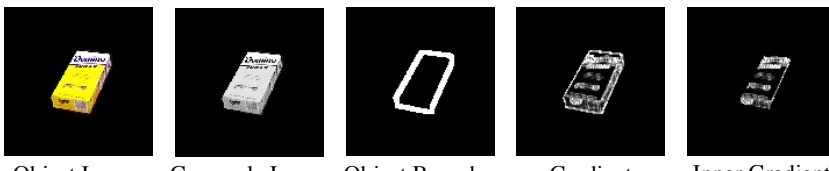

Object Image    Grayscale Image    Object Boundary    Gradient    Inner Gradient

Figure 11: The calculation of Object Color Gradient.

- **Object Shape Concavity:** As shown in Figure 12, given the binary mask of an object ($\boldsymbol{M}_{obj} \in \mathbb{R}^{H \times W}$), we first calculate its smallest surrounding convex polygon mask ($\boldsymbol{M}_{cvx} \in \mathbb{R}^{H \times W}$) using the existing algorithm [19]. Basically, this polygon mask is the smallest region that can cover any lines between two points on the original object mask. The object shape concavity factor is calculated as $1 - \sum \boldsymbol{M}_{obj} / \sum \boldsymbol{M}_{cvx}$. This factor naturally takes a value between 0 and 1. No further normalization is needed.

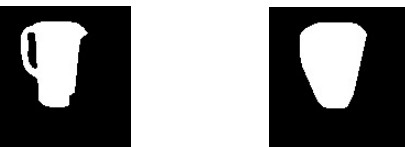

(a) Object Mask    (b) Smallest Convex Polygon Mask

Figure 12: An illustration of the Object Shape Concavity.

- **Inter-object Color Similarity:** As shown in Figure 13, we first calculate the average RGB color of each object. Each color corresponds to a point in the RGB space. We then calculate the Euclidean distance between each pair of object colors. The average value of all pairwise distances is divided by $255 \times \sqrt{3}$, which is the largest distance between two colors in the RGB space, so as to be normalized to the range of $[0, 1]$. The final score for inter-object color similarity is calculated as $1 - normalized\ RGB\ distance$.

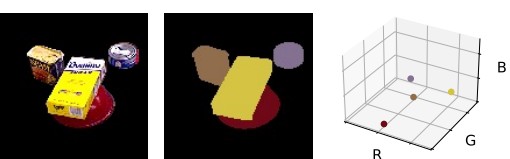

Figure 13: An illustration of the Inter-object Color Similarity.

- **Inter-object Shape Variation:** We first compute an axis-aligned bounding box for each object, and calculate its diagonal length. Then the differences of diagonal lengths between each pair of objects are calculated. The average value of such pairwise differences is the raw value of inter-object shape variation. This raw value is divided by $255 \times \sqrt{2}$, which is the largest possible difference of diagonal length, so as to be normalized to the range of $[0, 1]$.

## A.2    Other Candidates of Complexity Factors

In addition to the primary four complexity factors, we also explore other potential complexity factors to quantitatively measure the distributions of object- and scene-level biases in appearance and geometry. Basically, we aim to consider as many aspects as possible to investigate key factors underlying the distribution gaps between synthetic and real-world datasets. However, we empirically find that these candidate factors do not show significant discrepancy between synthetic and real-world datasets. Details are shown below.

### A.2.1    Candidates of Object-level Complexity Factors

- **Object Color Count:** This factor is defined as the total number of unique colors within an object mask. Basically, this is to simply measure the diversity of object color.
- **Object Color Entropy:** Inspired by Shannon entropy [49], we calculate the entropy value at each pixel by applying a $3 \times 3$ filter on the grayscale image concerted from RGB. In particular, for each pixel, its color value becomes a discrete value in $[0, 255]$. We compute its entropy score: $H(x) = -\sum_{i=1}^{n} p(x_i) \log_2 p(x_i)$, where $p(x_i)$ denotes the probability of a specific color value $x_i$ within the $3 \times 3$ neighbourhood. Basically, this factor aims to measure the color diversity within $3 \times 3$ image patches. The higher this factor, the more frequently the object color changes in small local areas.
- **Object Shape Non-rectangularity:** Given the binary mask of an object ($\boldsymbol{M}_{obj} \in \mathbb{R}^{H \times W}$), we first calculate its axis-aligned bounding box ($\boldsymbol{M}_{bbox} \in \mathbb{R}^{H \times W}$). Object shape non-rectangularity is calculated as $1 - \sum \boldsymbol{M}_{obj} / \sum \boldsymbol{M}_{bbox}$. Similar to object shape concavity, this factor is also designed to measure the complexity of object shapes. However, this factor is more likely to be affected by the object orientation since it takes axis-aligned bounding boxes as reference.
- **Object Shape Incompactness:** There are two similar methods to quantify the compactness of object shapes. The first one is Polsby–Popper test [44]: $PP(\boldsymbol{M}_{obj}) = 4\pi A(\boldsymbol{M}_{obj}) / P(\boldsymbol{M}_{obj})^2$. The other is Schwartzberg [48] compactness score: $S(\boldsymbol{M}_{obj}) = (2\pi \sqrt{A(\boldsymbol{M}_{obj})/\pi}) / P(\boldsymbol{M}_{obj})$. In both formula, $P(\boldsymbol{M}_{obj})$ is the object perimeter and $A(\boldsymbol{M}_{obj})$ is the object area. For simplicity, we choose $PP(\boldsymbol{M}_{obj})$ to calculate the object shape incompactness score: $1 - PP(\boldsymbol{M}_{obj})$.
- **Object Shape Discontinuity:** Given an object mask ($\boldsymbol{M}_{obj} \in \mathbb{R}^{H \times W}$), we first find the largest connected component ($\boldsymbol{M}_{lcc} \in \mathbb{R}^{H \times W}$) in its binary mask. The discontinuity of shape is calculated as: $1 - \sum \boldsymbol{M}_{lcc} / \sum \boldsymbol{M}_{obj}$. This factor is to evaluate how continuous an object shape is.
- **Object Shape Decentralization:** Given an object mask, we first calculate its centroid $(\bar{x}, \bar{y})$ by averaging all pixel coordinates in the object. Then, the second moment of this object is calculated as: $\sum_x \sum_y (x - \bar{x})^2 (y - \bar{y})^2$, where $(x, y)$ is the coordinates of pixels within the object. The higher this factor, the object shape is less likely to be centralized.

As shown in Figure 14, we compare the distributions of the object-level factor candidates on both synthetic and real-world datasets. It can be seen that the majority of these factors do not show significant gaps between the simple synthetic and the challenging real-world datasets. Therefore, we do not conduct relevant ablation experiments.

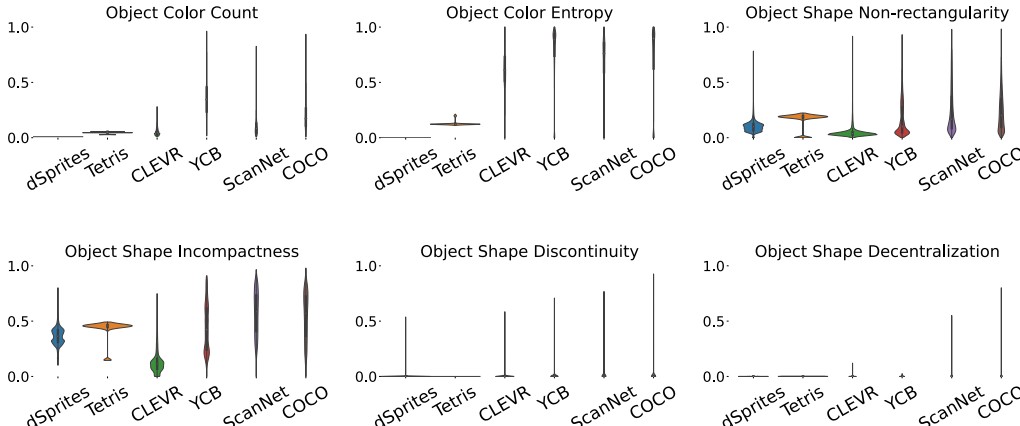

Figure 14: Distributions of Object-level Complexity Factors candidates.

### A.2.2 Candidates of Scene-level Complexity Factors

- **Inter-object Color Similarity with Chamfer Distance:** In the calculation of this factor, we first convert each pixel into a point in RGB space. In this way, each object can be represented by a point set in the RGB space. This factor is calculated between each pair of objects by measuring the Chamfer distance of two point sets in the RGB space. Since Chamfer distance is an asymmetric measurement, we calculate and average out the bidirectional Chamfer distances. Compared with Euclidean distance, this measurement favors the most similar colors between two objects.
- **Inter-object Color Similarity with Hausdorff Distance:** This factor is similar to the previous one. The only difference is that we replace Chamfer distance with Hausdorff distance. Hausdorff distance is also a directed and asymmetric measurement, so the final score is the average of distance values in both direction.
- **Inter-object Shape Similarity over Boundaries:** For each object mask, we first find its boundary using the method in [14], and then crop it with its axis-aligned bounding box. Each bounding box is scaled and fit into a unit box with its original aspect ratio. Lastly, we calculate the IoU between the boundaries of two objects to measure their shape similarity.
- **Inter-object Shape Entropy between Boundaries:** We first combine all object masks into a single image by assigning different indices to different objects. Then we compute the entropy of each pixel with a $3 \times 3$ filter. The final factor score is calculated by averaging all non-zero entropy values. Note that, the interior part of objects and background will not be considered because their entropy values will always be zeros. Basically, this factor is designed to evaluate how crowded an image is. The higher this factor, the more objects are spatially adjacent.
- **Inter-object Proximity between Centroids:** We first calculate the centroid $(\bar{x}, \bar{y})$ of each object by averaging all pixel coordinates in the object mask. Euclidean distances between object centroids are then computed pair-wisely before they are averaged to be the final factor score. This factor is designed to measure the spatial proximity of multiple objects in a single image.
- **Inter-object Proximity with Chamfer Distance:** In order to measure the spatial proximity between objects, we also calculate the spatial Chamfer distance between objects. Specifically, each object is represented by a set of $x - y$ coordinates, and then the average of pair-wise bidirectional Chamfer distances is calculated as the proximity score for each image.

As shown in Figure 15, we compare the distributions of the scene-level factor candidates on both synthetic and real-world datasets. We can see that both the inter-object color similarity with Chamfer and Hausdorff distances share similar distributions gaps with our primary inter-object color similarity factor defined in Section 2. The remaining four candidate factors relating to inter-object shape complexity do not show significant distribution gaps between the synthetic and real-world datasets. In this regard, we choose not to conduct ablation experiments on these six candidate factors.

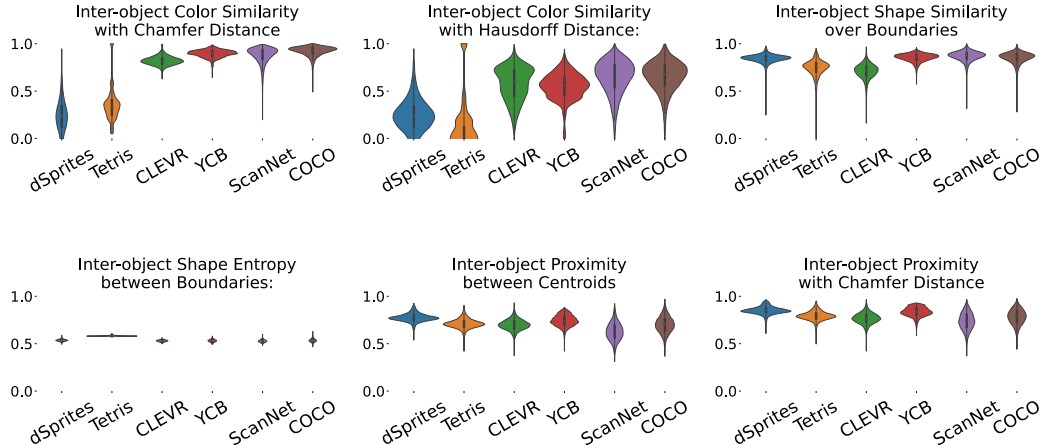

Figure 15: Distributions of Scene-level Complexity Factors candidates.

## A.3 Implementation Details

In this section, we present the implementation details of four representative models.

### AIR [22]

- **Source Code:** We refer to https://pyro.ai/examples/air.html and https://github.com/addtt/attend-infer-repeat-pytorch for the implementation.
- **Important Adaptations:** We use an additional parameter to weight the KL divergence loss and reconstruction loss. For each experiment, we choose the weight for KL divergence from 1, 10, 25 and 50. The highest AP score is kept.
- **Training Details:** All experiments of AIR [22] are conducted with a batch size of 64. The learning rate is set to $1e-4$ for training inference networks and decoders, $1e-3$ for baselines which is the same as the original paper. Since the number of objects in our datasets ranges between 2 and 6, we set the maximum number of steps at inference to be 6 for all experiments. All models are trained on a single GPU for 1000 epochs. We perform evaluation every 50 epochs and select the one with the highest AP score.

### MONet [8]

- **Source Code:** We refer to [21]'s re-implementation at: https://github.com/applied-ai-lab/genesis.
- **Important Adaptations:** We train MONet [8] with the GECO objective [46] following the protocol mentioned in [21].
- **Training Details:** All experiments on MONet [8] are conducted with a batch size of 32 and learning rate of $1e-4$. Since the maximum number of components is 7, including 1 background and 6 objects, we set the number of steps to be 7 for all experiments. All models are trained on a single GPU for 200 epochs with the training loss converged. We perform evaluation every 10 epochs and select the one with the highest AP score.

### IODINE [25]

- **Source Code:** We use the official implementation at: https://github.com/deepmind/deepmind-research/tree/master/iodine .
- **Important Adaptations:** The architecture is set the same as what is used for CLEVR dataset [33] in the original paper [25].
- **Training Details:** Since we use a single GPU for the training of all models, the batch size is adjusted to be 4 and learning rate $0.0001 \times \sqrt{1/8}$. The number of slots $K$ is set as 7 and the inference iteration $T$ as 5. We train each model for $500K$ iterations until the loss is fully converged.

### SlotAtt [40]

- **Source Code:** We use the official implementation at: https://github.com/google-research/google-research/tree/master/slot_attention.
- **Important Adaptations:** The architecture is set the same as what is used for CLEVR dataset [33] in the original paper [40].
- **Training Details:** All experiments of SlotAtt [40] are conducted with a batch size of 32 and learning rate selected from [$4e-4$, $4e-5$]. The number of slots $K$ is set as 7 and the number of iterations $T$ is set as 3. All models are trained on a single GPU for $500K$ iterations until the loss is fully converged.

### Mask R-CNN [29]

- **Source Code:** We use the implementation at: https://github.com/matterport/Mask_RCNN.
- **Important Adaptations:** We use the same settings as training for COCO in above repository.
- **Training Details:** Training for all datasets starts from the pre-trained COCO weights (mask_rcnn_coco.h5) from https://github.com/matterport/Mask_RCNN/releases. All models are trained on a single GPU for 30 epochs until the loss is fully converged.

## A.4 Details of six Benchmark Datasets

In this section, we present the details of three synthetic and three real-world datasets.

**dSprites [42]** To generate a specific image for this dataset, we first sample a random integer $K$ from a uniform distribution with interval $[2, 6]$ as the number of objects in that image. Then, $K$ object shapes are selected from the binary dsprites dataset [42] also in a uniformly random manner. Each object is assigned with a random RGB color by sampling three random integers from a uniform distribution with interval $[0, 255]$. In total, we generate 10000 images for training, 2000 for testing.

**Tetris [34]** For each image in this dataset, we first sample a random integer $K$ from a uniform distribution with interval $[2, 6]$ as the number of objects in this image. To render one tetris-like object onto the canvas, we randomly pick up a tetris object from a randomly selected image from [34]. Each object is resized to be $88 \times 88$ and then placed onto the canvas. The position of each object is also sampled from a uniform distribution with 2 criteria: 1) all objects shall be on the canvas with complete shapes; 2) all objects shall not overlap with each other.

**CLEVR [33]** We first generate CLEVR images following https://github.com/facebookresearch/clevr-dataset-gen, where the number of objects per image is restricted between 3 to 6. Given generated images with a resolution $640 \times 480$, we perform center-cropping and then resize them to be $128 \times 128$. Then, we remove tiny objects which have less than 35 pixels from each image. Subsequently, the images with less than 2 objects are removed. Being consistent with previous 2 synthetic datasets, all images have a black background.

**YCB [9]** We sample single frames from the YCB video dataset [9] every 20 images. Given the sampled frames with a resolution of $640 \times 480$, we first center crop and then resize them to be $128 \times 128$. Then, the images consisting of less than 2 or more than 6 objects are removed. Similarly, all background pixels are replaced by a black color.

**ScanNet [17]** We sample single frames from the ScanNet dataset [17] every 20 images. Given the selected frames with a resolution of $1296 \times 968$, we first center crop the images with a size of $800 \times 800$ and then resize them to be $128 \times 128$. For each resized image, we remove objects that contain more than $128 \times 128 \times 0.2$ pixels or less than $128 \times 128 \times 0.007$ pixels. The images with less than 2 or more than 6 are also dropped. All background pixels are replaced by the black color.

**COCO [38]** Given images in COCO-2017 [38] with various resolutions, we first center crop and then resize images to be $128 \times 128$. For each resized image, we use the same criteria applied to ScanNet to remove too large and too small objects. The images with $2 \sim 6$ objects are kept. All background pixels are replaced by the black color. Since the number of images that could meet all requirements is less than 2,000 (only 1,597) in the official validation split, we additionally select 403 different images from its official training split for testing. There is no overlap between our training and testing splits.

Figure 16 shows three example images for each dataset. The distribution of images with different number of objects is also presented.

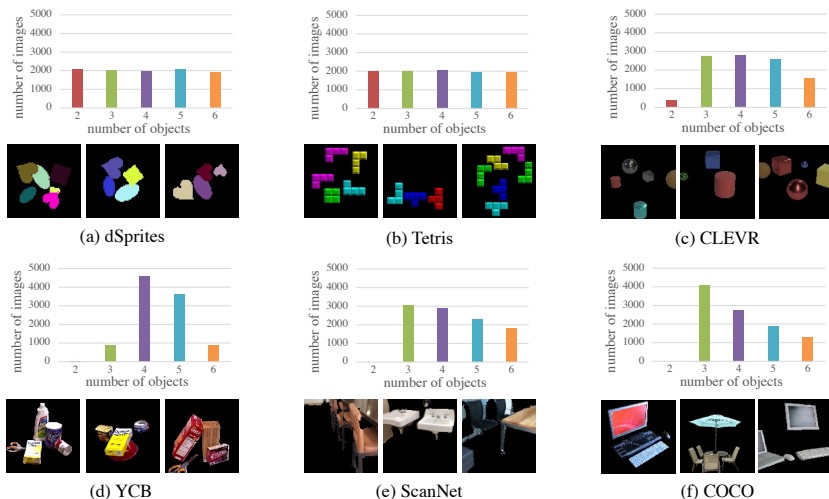

Figure 16: Example images and the statistics of the six benchmark datasets.

## A.5 Details and Results of Main Experiments

In this section, we present the result of four unsupervised methods and one supervised method on six datasets. Both quantitative evaluation and qualitative results are provided.

### A.5.1 Four Unsupervised Methods

Figure 17 shows the qualitative results of 4 representative methods on 6 datasets. The results of AIR [22] are presented with predicted bounding boxes while others are represented with predicted segmentation masks. Different colors in a segmentation mask indicate different objects. For the same object, the assigned color may not be the same. We can see that all methods give reasonable results on the three synthetic datasets in spite of some performance gaps between them due to the different capability of these methods. However, they all fail on the three real-world datasets. Specifically, AIR [22] cannot reconstruct reasonable images. MONet [8] always performs segmentation based on color. IODINE [25] cannot recover meaningful object shapes and locations. SlotAtt [40] can roughly locate objects but cannot identify the real object shapes.

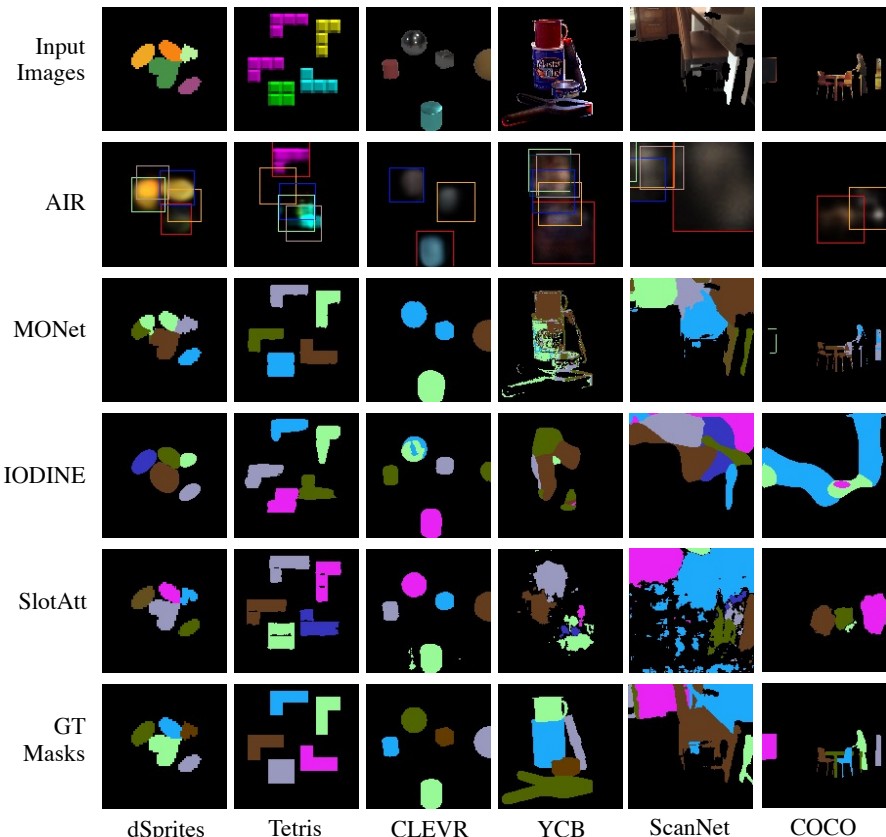

Figure 17: Qualitative results of object segmentation from the four methods on six datasets.

Table 3: Quantitative results of object segmentation from the four methods on six datasets. Standard deviations of performance are calculated over 3 runs (marked with blue).

| | dSprites | Tetris | CLEVR |
|---|---|---|---|
| | AP / PQ / Pre / Rec | AP / PQ / Pre / Rec | AP / PQ / Pre / Rec |
| AIR [22] | 45.4 1.8 / 38.2 3.0 / 57.6 7.4 / 58.1 7.5 | 25.2 13.9 / 23.4 12.4 / 36.8 20.9 / 39.9 12.9 | 46.4 14.0 / 44.3 12.4 / 67.4 9.9 / 52.5 15.9 |
| MONet [8] | 69.7 4.1 / 61.6 6.0 / 70.4 8.1 / 73.9 1.9 | 85.9 13.0 / 75.8 13.6 / 85.1 16.4 / 89.7 8.2 | 39.0 8.5 / 37.3 6.3 / 65.6 11.8 / 42.8 10.8 |
| IODINE [25] | 92.9 4.3 / 71.3 6.1 / 82.6 2.3 / 96.0 5.2 | 52.2 2.3 / 37.9 4.6 / 48.0 2.3 / 61.8 1.7 | 82.8 2.8 / 73.0 5.7 / 77.5 3.1 / 87.4 2.0 |
| SlotAtt [40] | 92.8 1.4 / 82.8 1.6 / 88.8 3.4 / 92.9 1.6 | 94.3 1.2 / 79.9 6.4 / 90.5 3.3 / 94.4 1.3 | 91.7 6.4 / 82.9 10.9 / 90.8 9.7 / 92.7 5.3 |

| | YCB | ScanNet | COCO |
|---|---|---|---|
| | AP / PQ / Pre / Rec | AP / PQ / Pre / Rec | AP / PQ / Pre / Rec |
| AIR [22] | 0.0 0.1 / 0.6 0.3 / 1.1 0.4 / 0.8 0.2 | 2.7 1.4 / 6.3 1.7 / 15.6 2.8 / 7.3 1.6 | 2.7 0.1 / 6.7 0.5 / 14.3 2.6 / 8.6 0.8 |
| MONet [8] | 3.1 1.6 / 7.0 2.6 / 9.8 3.6 / 1.2 0.8 | 24.8 1.6 / 24.6 1.6 / 31.0 1.6 / 40.7 1.8 | 11.8 2.0 / 12.5 1.1 / 16.1 0.9 / 21.9 1.7 |
| IODINE [25] | 1.8 0.2 / 3.9 1.3 / 6.2 2.0 / 7.3 1.9 | 10.1 2.9 / 13.7 2.7 / 18.6 4.2 / 24.4 3.8 | 4.0 1.2 / 6.3 1.2 / 9.9 1.8 / 10.8 2.0 |
| SlotAtt [40] | 9.2 0.4 / 13.5 0.9 / 20.0 1.3 / 26.2 6.8 | 5.7 0.3 / 9.0 1.5 / 12.4 2.5 / 18.3 2.7 | 0.8 0.3 / 3.5 1.2 / 5.3 1.7 / 7.3 2.2 |

### A.5.2 One Supervised Method

Apart from above four unsupervised approaches, we also include Mask R-CNN [29] as a supervised object segmentation baseline. As shown in Figure 18, Mask R-CNN exhibits quite successful segmentation on three synthetic datasets. Real-world datasets are more challenging for Mask R-CNN, but the results are much better than unsupervised baselines.

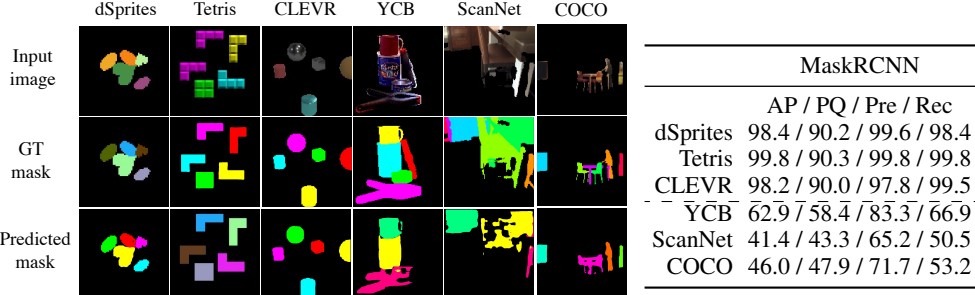

| | MaskRCNN |
| --- | --- |
| | AP / PQ / Pre / Rec |
| dSprites | 98.4 / 90.2 / 99.6 / 98.4 |
| Tetris | 99.8 / 90.3 / 99.8 / 99.8 |
| CLEVR | 98.2 / 90.0 / 97.8 / 99.5 |
| YCB | 62.9 / 58.4 / 83.3 / 66.9 |
| ScanNet | 41.4 / 43.3 / 65.2 / 50.5 |
| COCO | 46.0 / 47.9 / 71.7 / 53.2 |

Figure 18: Qualitative and quantitative results of Mask-RCNN on six datasets.

## A.6 Details and Results of Ablation Experiments

In this section, we shows details of ablated datasets including example images. Both qualitative and quantitative evaluation are also presented. Table 4 is a look-up table for individual ablated factor.

Table 4: A look-up table for ablations.

| | Which level | | Which aspect | | Target factor | | | |
| --- | --- | --- | --- | --- | --- | --- | --- | --- |
| Ablation | Object | Scene | Color | Shape | Object Color Gradient | Object Shape Concavity | Inter-object Color Similarity | Inter-object Shape Variation |
| C | ✓ | | ✓ | | ✓ | | | |
| S | ✓ | | | ✓ | | ✓ | | |
| T | | ✓ | ✓ | | | | ✓ | |
| U | | ✓ | | ✓ | | | | ✓ |

### A.6.1 Ablations on Object-level Factors

**Ablated Datasets**

- *Ablation of Object Color Gradient*: In the three ablated datasets: YCB-C / ScanNet-C / COCO-C, each object is represented by a single average color.
- *Ablation of Object Shape Concavity*: In the three ablated datasets: YCB-S / ScanNet-S / COCO-S, each object is represented by a convex shape.
- *Ablation of both Object Color Gradient and Shape Concavity*: In the three ablated datasets: YCB-C+S / ScanNet-C+S / COCO-C+S, each object is represented by a convex shape with a single average color.

Figure 19 shows example images from the three real-world datasets with different types of object-level factor ablations.

**Qualitative and Quantitative Results**    As shown in Figure 20 and Table 5, all four methods have a significant improvement in segmentation performance on the ablated datasets with object color gradients being removed. By comparison, the ablation of object shape concavity is less effective. This shows that both object-level factors are relevant to the success of object segmentation, although object color gradient is more important for the four methods. Specifically, MONet [8] and IODINE [25] are more sensitive to color gradient compared with the other two methods.

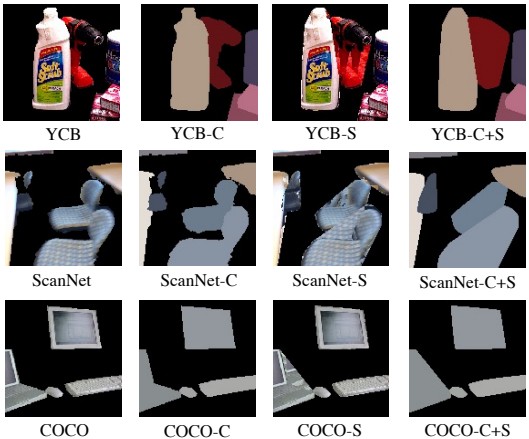

Figure 19: Example images of real-world datasets ablated with object-level factors.

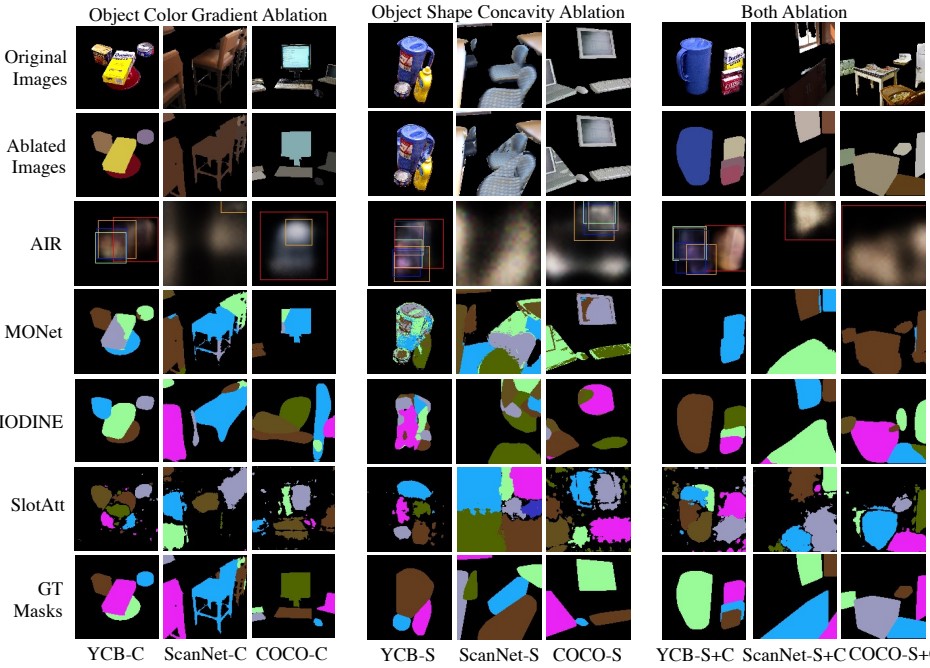

Figure 20: Qualitative results on the datasets ablated with object-level factors.

Table 5: Quantitative results on the datasets ablated with object-level factors. Standard deviations of performance are calculated over 3 runs (marked with blue).

| | YCB-C | ScanNet-C | COCO-C |
|---|---|---|---|
| | AP / PQ / Pre / Rec | AP / PQ / Pre / Rec | AP / PQ / Pre / Rec |
| AIR [22] | 4.4 0.2 / 1.0 7.3 / 20.7 7.1 / 12.5 1.2 | 2.7 1.0 / 7.6 2.1 / 29.1 20.2 / 7.4 1.5 | 2.9 1.8 / 7.9 2.5 / 32.0 29.3 / 7.7 4.5 |
| MONet [8] | 55.1 6.0 / 49.5 5.2 / 64.8 1.7 / 58.8 7.8 | 31.1 15.9 / 33.1 9.6 / 41.6 10.3 / 45.6 11.1 | 34.5 7.2 / 34.6 3.3 / 45.8 2.0 / 42.6 11.0 |
| IODINE [25] | 76.8 0.3 / 64.4 0.1 / 76.5 0.1 / 79.5 0.0 | 64.4 6.4 / 54.3 5.9 / 66.3 4.6 / 71.0 5.9 | 42.8 14.7 / 34.0 11.8 / 42.5 14.4 / 55.8 11.9 |
| SlotAtt [40] | 39.2 1.9 / 30.2 0.0 / 40.2 0.8 / 50.4 0.7 | 5.7 8.2 / 9.0 6.1 / 12.4 7.6 / 18.3 8.9 | 7.2 1.9 / 9.8 1.7 / 13.9 2.2 / 18.7 3.4 |

| | YCB-S | ScanNet-S | COCO-S |
|---|---|---|---|
| | AP / PQ / Pre / Rec | AP / PQ / Pre / Rec | AP / PQ / Pre / Rec |
| AIR [22] | 3.6 3.0 / 6.8 3.2 / 12.5 6.7 / 9.7 2.7 | 3.3 2.5 / 8.5 1.1 / 36.6 23.0 / 8.1 8.5 | 3.3 1.0 / 16.8 5.6 / 8.3 1.0 |
| MONet [8] | 2.5 1.1 / 6.4 1.8 / 8.9 2.6 / 11.3 3.2 | 18.3 10.4 / 22.2 3.9 / 28.4 4.4 / 37.5 5.3 | 8.7 3.2 / 11.2 1.7 / 14.4 3.0 / 21.0 0.7 |
| IODINE [25] | 2.5 2.5 / 5.0 5.0 / 7.9 7.8 / 1.1 1.0 | 8.3 1.1 / 13.7 0.4 / 18.4 0.4 / 24.8 0.7 | 4.2 0.4 / 8.3 0.4 / 12.1 0.3 / 14.8 0.6 |
| SlotAtt [40] | 19.1 0.7 / 21.0 1.7 / 30.4 2.8 / 37.7 0.8 | 1.9 8.5 / 6.6 7.5 / 8.7 9.8 / 13.7 11.9 | 2.6 1.0 / 5.9 0.9 / 8.3 1.3 / 11.9 10.9 |

| | YCB-C+S | ScanNet-C+S | COCO-C+S |
|---|---|---|---|
| | AP / PQ / Pre / Rec | AP / PQ / Pre / Rec | AP / PQ / Pre / Rec |
| AIR [22] | 2.0 0.3 / 6.4 0.2 / 15.1 5.2 / 7.3 4.1 | 3.5 1.0 / 9.1 0.5 / 38.3 25.8 / 8.5 5.9 | 4.8 0.1 / 10.8 0.1 / 41.5 10.2 / 10.1 1.7 |
| MONet [8] | 9.1 13.6 / 12.7 10.5 / 34.1 4.6 / 12.6 24.0 | 48.1 1.0 / 47.9 6.8 / 70.4 23.3 / 51.5 6.5 | 10.5 24.3 / 16.5 18.1 / 56.5 6.9 / 14.9 28.2 |
| IODINE [25] | 69.9 30.9 / 55.7 29.6 / 60.3 28.3 / 71.5 26.4 | 73.8 11.4 / 63.2 10.1 / 73.5 6.1 / 76.8 10.8 | 73.5 2.5 / 61.2 2.4 / 70.0 2.9 / 77.6 1.2 |
| SlotAtt [40] | 39.1 22.3 / 30.2 13.3 / 42.9 12.2 / 54.5 12.7 | 22.5 7.6 / 21.5 6.2 / 28.8 6.0 / 39.6 5.4 | 20.3 3.0 / 19.9 0.4 / 26.1 0.4 / 34.8 0.7 |

### A.6.2    Ablations on Scene-level Factors

**Ablated Datasets**

- *Ablation of Inter-object Color Similarity*: In the three ablated datasets: YCB-T / ScanNet-T / COCO-T, each object's original texture is replaced by one of the selected 6 distinctive textures from the DTD database [15], as shown in Figure 21. In particular, the 6 textures are chosen from the 'blotchy' category, and we deliberately select those with distinctive colors. The texture images are center-cropped and resized to a resolution of $128 \times 128$. Each object appearance is replaced by one of the 6 texture images, and each object in a single image is assigned with a different texture.
- *Ablation of Inter-object Shape Variation*: In the three ablated datasets: YCB-U / ScanNet-U / COCO-U, the size of each object is uniformly scaled. In particular, before generating the ablated datasets, we first calculate the average scale of objects in each dataset by calculating the mean of object bounding box diagonals (YCB: 60 / ScanNet: 70 / COCO: 57, in pixels). Each object in a specific dataset is scaled up or down to approach the average scale in this dataset. The shape and appearance are also linearly scaled, so that the shape variation is the only factor changed in the new dataset.
- *Ablation of both Inter-object Color Similarity and Shape Variation*: In the three ablated datasets: YCB-T+U / ScanNet-T+U / COCO-T+U, each object's texture is replaced and shape is uniformly scaled.

Figure 22 shows example images from the three real-world datasets with different types of scene-level factor ablations.

**Qualitative and Quantitative Results**    As shown in Figure 23 and Table 6, all methods have a significant improvement in object segmentation on the ablated datasets with reduced inter-object color similarity. The effect of uniform scale ablation is limited standalone. However, when both ablations are combined, there is a larger performance gain compared with purely texture replaced ablation. Specifically, AIR [22] is sensitive to both scene-level factors, since only the combination of two factors can result in a significant improvement. MONet [8] and IODINE [25], however, are almost only sensitive to the inter-object color similarity. Both factors have a noticeable effect on the segmentation performance of SlotAtt [40].

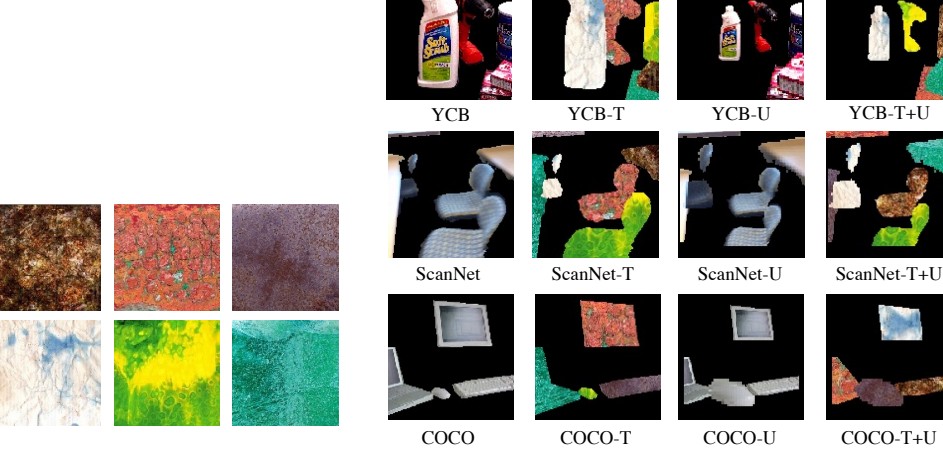

Figure 21: Six selected texture images from DTD [15].

Figure 22: Example images of datasets ablated with scene-level factors.

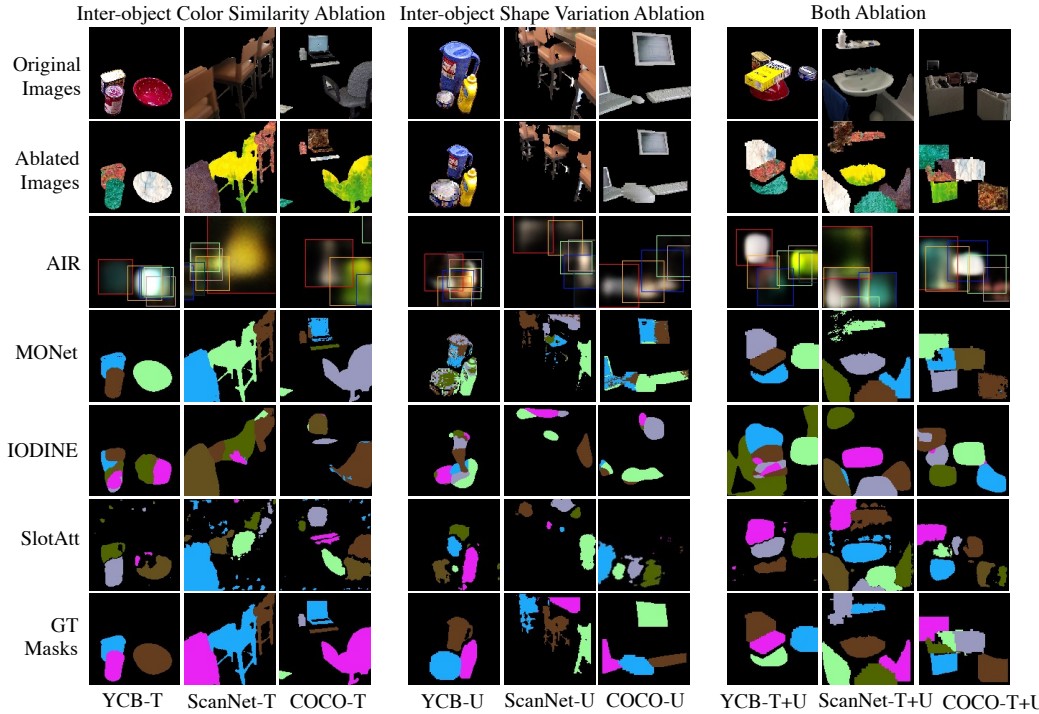

Figure 23: Qualitative results on the datasets ablated with scene-level factors.

Table 6: Quantitative results on the datasets ablated with scene-level factors. Standard deviations of performance are calculated over 3 runs (marked with blue).

| | YCB-T | ScanNet-T | COCO-T |
|---|---|---|---|
| | AP / PQ / Pre / Rec | AP / PQ / Pre / Rec | AP / PQ / Pre / Rec |
| AIR [22] | 5.9 3.7 / 9.6 3.3 / 19.9 10.3 / 12.7 1.3 | 2.9 1.2 / 6.3 0.1 / 13.4 3.7 / 8.6 3.2 | 7.4 3.1 / 13.0 4.0 / 29.1 9.8 / 16.4 4.3 |
| MONet [8] | 86.5 12.4 / 78.3 13.3 / 81.1 11.6 / 86.8 12.2 | 82.9 12.3 / 77.8 7.6 / 86.3 1.6 / 83.7 11.9 | 80.0 9.4 / 68.8 13.5 / 74.5 15.0 / 82.0 5.6 |
| IODINE [25] | 32.4 9.0 / 27.3 6.8 / 35.3 8.3 / 43.6 10.6 | 33.2 11.9 / 27.3 6.3 / 34.5 7.0 / 44.7 9.3 | 40.8 12.7 / 33.7 10.2 / 41.8 **10.4** / 55.9 15.8 |
| SlotAtt [40] | 64.6 5.3 / 48.5 3.2 / 58.3 3.5 / 74.9 4.6 | 20.5 22.3 / 18.2 20.1 / 22.9 23.1 / 33.2 31.4 | 31.8 17.8 / 28.0 11.2 / 35.2 12.4 / 50.1 19.8 |

| | YCB-U | ScanNet-U | COCO-U |
|---|---|---|---|
| | AP / PQ / Pre / Rec | AP / PQ / Pre / Rec | AP / PQ / Pre / Rec |
| AIR [22] | 9.8 3.5 / 12.2 2.6 / 18.6 4.1 / 19.9 2.8 | 7.1 1.4 / 10.4 0.2 / 19.9 2.1 / 14.0 1.0 | 12.9 4.8 / 16.3 4.6 / 28.6 1.8 / 24.3 9.3 |
| MONet [8] | 3.7 0.5 / 8.3 1.2 / 12.2 1.4 / 13.2 1.4 | 25.4 8.3 / 23.6 7.4 / 29.2 8.9 / 37.2 12.5 | 14.5 0.6 / 16.5 0.8 / 27.9 1.4 / 20.9 0.8 |
| IODINE [25] | 2.5 1.4 / 4.9 2.0 / 7.6 2.9 / 9.3 3.6 | 16.9 0.8 / 17.2 0.0 / 23.8 0.8 / 30.2 0.1 | 5.5 1.0 / 7.7 0.0 / 11.7 0.6 / 13.5 0.3 |
| SlotAtt [40] | 28.9 2.0 / 21.9 1.2 / 30.7 3.1 / 37.3 2.4 | 9.4 4.8 / 10.4 3.8 / 15.4 4.6 / 18.1 7.2 | 4.4 1.1 / 7.9 0.3 / 11.9 0.9 / 14.7 0.7 |

| | YCB-T+U | ScanNet-T+U | COCO-T+U |
|---|---|---|---|
| | AP / PQ / Pre / Rec | AP / PQ / Pre / Rec | AP / PQ / Pre / Rec |
| AIR [22] | 18.9 13.7 / 21.0 9.9 / 32.5 15.7 / 32.6 13.2 | 12.3 5.4 / 16.8 5.6 / 37.4 20.8 / 20.4 1.1 | 25.3 13.0 / 28.3 9.0 / 49.4 13.2 / 38.2 12.4 |
| MONet [8] | 89.1 0.8 / 84.7 1.5 / 91.7 4.1 / 89.3 0.8 | 87.0 8.9 / 77.5 10.3 / 80.5 11.1 / 87.4 8.7 | 88.2 2.9 / 76.2 0.1 / 79.9 3.7 / 89.0 2.9 |
| IODINE [25] | 35.3 1.2 / 26.4 0.3 / 34.9 0.6 / 42.8 0.2 | 27.4 0.8 / 25.0 0.8 / 32.1 0.5 / 41.8 1.8 | 53.3 16.7 / 35.6 8.3 / 45.0 9.7 / 59.8 13.0 |
| SlotAtt [40] | 76.8 1.2 / 57.1 3.8 / 72.4 3.4 / 77.7 1.8 | 47.3 11.2 / 33.8 9.2 / 42.6 9.1 / 57.3 8.4 | 34.5 17.8 / 25.3 7.6 / 33.7 8.3 / 43.0 14.1 |

### A.6.3 Ablations on Object- and Scene-Level Factor Joint Ablations - Part 1

**Ablated Datasets**

- *Ablation of Object Color Gradient, Object Shape Concavity and Inter-object Color Similarity*: In the three ablated datasets: YCB-C+S+T / ScanNet-C+S+T / COCO-C+S+T, for each image, we first find the smallest convex hull [19] for each object. Then each convex hull is filled with a distinctive texture as shown in Figure 21. Lastly, we calculate the average color of the distinctive texture in each convex hull and change each pixel inside to be that color.

- *Ablation of Object Color Gradient, Object Shape Concavity and Inter-object Shape Variation*: In the three ablated datasets: YCB-C+S+U / ScanNet-C+S+U / COCO-C+S+U, we use the (C+S) datasets created before and find the uniform scale of the object convex hull.

- *Ablation of all four factors*: In the three ablated datasets: YCB-C+S+T+U / ScanNet-C+S+T+U / COCO-C+S+T+U, each image is ablated with the four factors together.

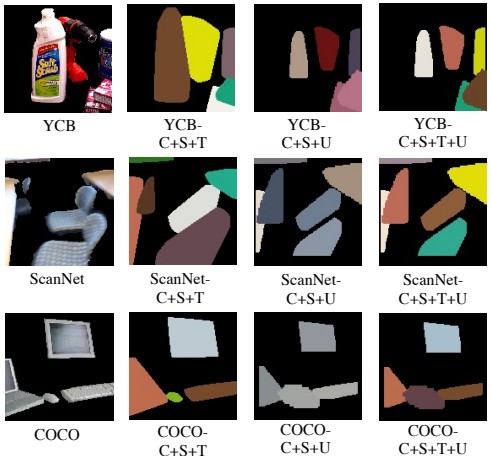

| YCB | YCB-
C+S+T | YCB-
C+S+U | YCB-
C+S+T+U |
| ScanNet | ScanNet-
C+S+T | ScanNet-
C+S+U | ScanNet-
C+S+T+U |
| COCO | COCO-
C+S+T | COCO-
C+S+U | COCO-
C+S+T+U |

Figure 24: Example images of datasets ablated with both object- and scene-level factors.

Figure 24 shows example images from the three real-world datasets with different types of joint object- and scene-level factor ablations.

**Qualitative and Quantitative Results**  As shown in Figure 25 and Table 7, all methods have achieved a significant improvement in object segmentation on the datasets ablated with joint object- and scene-level factors. The datasets with all four factors ablated can lead to impressive performance similar to the synthetic datasets. This shows that the distribution gaps of the objectness biases represented by the four factors between the synthetic and real-world datasets lead to the failure of existing unsupervised models.

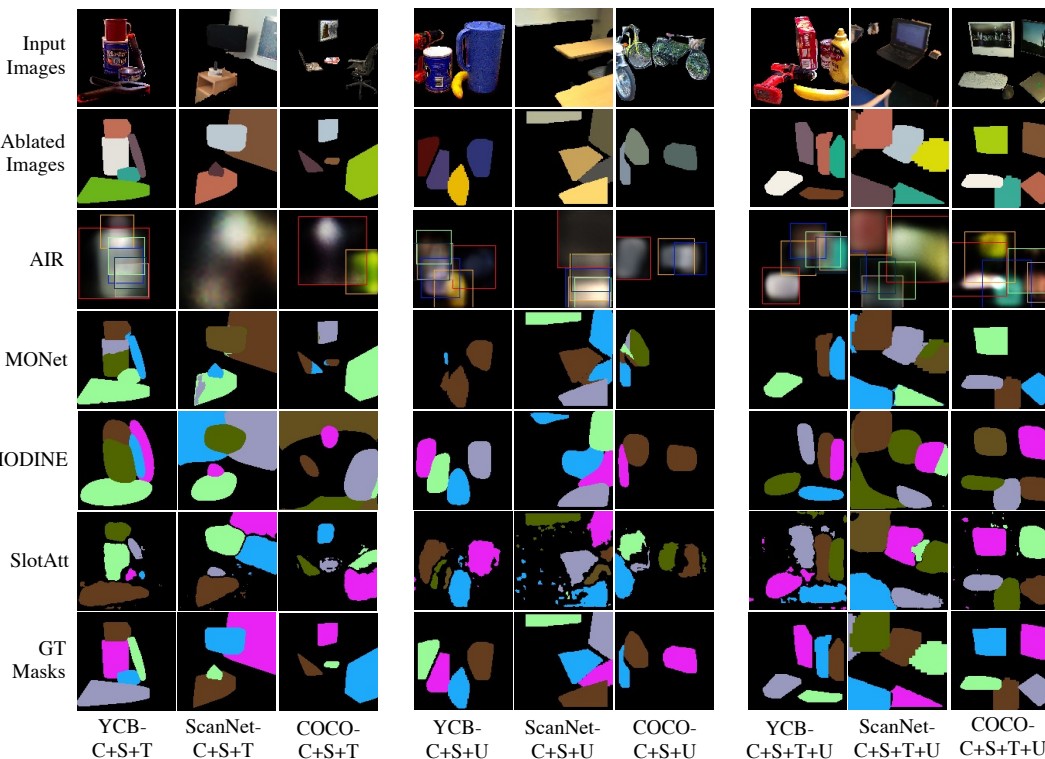

Figure 25: Qualitative results on the datasets ablated with both object- and scene-level factors.

Table 7: Quantitative results on the datasets ablated with both object- and scene-level factors. Standard deviations of performance are calculated over 3 runs (marked with blue).

| | YCB-C+S+T | ScanNet-C+S+T | COCO-C+S+T |
|---|---|---|---|
| | AP / PQ / Pre / Rec | AP / PQ / Pre / Rec | AP / PQ / Pre / Rec |
| AIR [22] | 6.3 3.7 / 10.7 3.3 / 24.0 13.2 / 13.1 0.5 | 1.9 3.9 / 6.4 5.2 / 27.6 10.7 / 6.3 13.9 | 10.8 5.6 / 18.6 7.0 / 45.1 14.4 / 20.3 6.9 |
| MONet [8] | 67.7 4.2 / 55.0 8.5 / 62.7 19.8 / 75.4 10.2 | 76.1 7.4 / 61.7 2.6 / 64.4 2.1 / 77.1 2.8 | 67.1 15.3 / 56.4 10.1 / 62.2 6.2 / 73.9 17.3 |
| IODINE [25] | 82.6 2.6 / 63.8 0.9 / 75.5 5.1 / 86.0 1.6 | 69.0 14.9 / 53.5 14.1 / 60.7 14.5 / 77.3 9.4 | 77.6 7.8 / 59.8 6.8 / 72.5 1.8 / 83.0 7.5 |
| SlotAtt [40] | 65.9 23.2 / 51.6 18.5 / 62.2 19.8 / 72.9 17.6 | 60.2 8.7 / 45.5 8.1 / 52.9 9.2 / 68.3 7.3 | 51.8 11.0 / 41.5 7.7 / 48.6 7.5 / 62.9 8.5 |
| | YCB-C+S+U | ScanNet-C+S+U | COCO-C+S+U |
| | AP / PQ / Pre / Rec | AP / PQ / Pre / Rec | AP / PQ / Pre / Rec |
| AIR [22] | 12.5 6.4 / 17.7 7.0 / 27.5 11.2 / 29.6 10.5 | 8.2 1.5 / 11.1 0.5 / 20.4 0.8 / 15.0 1.3 | 23.7 9.8 / 27.5 8.0 / 51.3 9.6 / 32.8 8.3 |
| MONet [8] | 3.5 2.5 / 4.8 3.2 / 23.5 2.6 / 5.3 6.5 | 60.1 9,5 / 56.3 13.3 / 69.7 20.7 / 64.0 6.2 | 35.6 1.4 / 33.0 3.8 / 43.4 16.6 / 38.7 0.1 |
| IODINE [25] | 65.3 9.2 / 55.9 10.4 / 73.9 4.6 / 67.7 8.4 | 64.6 3.2 / 52.1 5.1 / 61.3 7.6 / 68.2 2.2 | 64.8 7.3 / 53.4 6.5 / 65.8 8.5 / 68.2 2.5 |
| SlotAtt [40] | 58.1 3.7 / 39.1 2.8 / 52.4 2.3 / 62.0 3.6 | 50.5 2.9 / 38.2 3.4 / 51.8 6.9 / 55.0 1.3 | 20.2 5.2 / 18.1 0.3 / 26.9 0.1 / 33.4 0.6 |
| | YCB-C+S+T+U | ScanNet-C+S+T+U | COCO-C+S+T+U |
| | AP / PQ / Pre / Rec | AP / PQ / Pre / Rec | AP / PQ / Pre / Rec |
| AIR [22] | 24.8 20.1 / 23.2 11.9 / 33.7 16.8 / 39.1 20.1 | 10.0 1.8 / 13.5 1.8/ 24.0 1.4 / 18.2 7.4 | 36.4 19.3 / 39.4 14.5 / 63.1 16.9 / 49.2 17.9 |
| MONet [8] | 70.8 9.5 / 79.2 13.0 / 96.8 9.4 / 70.8 8.5 | 79.2 3.4 / 65.0 3.6 / 69.1 4.2 / 82.4 2.2 | 76.2 3.0 / 75.9 3.8 / 90.9 6.4 / 76.5 3.3 |
| IODINE [25] | 72.4 4.8 / 54.2 6.7 / 65.1 5.8 / 74.3 5.8 | 76.1 8.7 / 56.5 9.2 / 66.9 7.0 / 80.3 6.2 | 81.4 3.4 / 59.3 5.1 / 70.5 7.7 / 84.3 6.7 |
| SlotAtt [40] | 92.0 2.0 / 65.5 6.1 / 84.4 6.5 / 92.5 1.7 | 62.7 25.4 / 42.6 28.4 / 50.5 34.3 / 69.4 19.0 | 83.7 4.7 / 60.7 7.2 / 76.4 4.6 / 84.0 5.0 |

### A.6.4 Ablations on Object- and Scene-Level Factor Joint Ablations - Part 2

In addition to the experiments in Section A.6.3, we generate additional 6 groups of datasets ablated with different combinations of both object- and scene-level factors as detailed in Section A.6.4, and conduct extra experiments shown in Section A.6.4. Figure 26 shows example images of the additional ablated datasets.

**Ablated Datasets**

- *Ablation of Object Color Gradient and Inter-object Color Similarity*: In each image of the three real-world datasets, we replace the object color by averaging all pixels of a distinctive texture, and keep the original shape unchanged, getting three ablated datasets: YCB-C+T / ScanNet-C+T / COCO-C+T.

- *Ablation of Object Color Gradient and Inter-object Shape Variation*: In each image of the three real-world datasets, we replace the object color by averaging its own texture, and then apply size normalization on the original shape of objects, getting three ablated datasets: YCB-C+U / ScanNet-C+U / COCO-C+U.

- *Ablation of Object Color Gradient, Inter-object Color Similarity and Inter-object Shape Variation*: In each image of the three real-world datasets, we replace the object color by averaging all pixels of a distinctive texture, and then apply size normalization on the original shape of objects. The ablated datasets are denoted as: YCB-C+T+U / ScanNet-C+T+U / COCO-C+T+U.

- *Ablation of Object Shape Concavity and Inter-object Color Similarity*: In each image of the three real-world datasets, we replace the object color by a distinctive texture, and modify the object shape as a convex hull, getting three ablated datasets: YCB-S+T / ScanNet-S+T / COCO-S+T.

- *Ablation of Object Shape Concavity and Inter-object Shape Variation*: In each image of the three real-world datasets, we keep the texture of object unchanged, and modify the object shape as a convex hull followed by size normalization, getting three ablated datasetss: YCB-S+U / ScanNet-S+U / COCO-S+U.

- *Ablation of Object Shape Concavity, Inter-object Color Similarity and Inter-object Shape Variation*: In each image of the three real-world datasets, we replace the object color by a distinctive texture, and modify the object shape as a convex hull followed by size normalization. The ablated datasets are denoted as: YCB-S+T+U / ScanNet-S+T+U / COCO-S+T+U.

**Qualitative and Quantitative Results** As shown in Table 8 and Figures 27/28/29/30, all 6 additional combinations of object- and scene-level factors are explored, demonstrating consistent findings as our experiments in Section 4.4. Overall, all four methods show a high sensitivity to both object- and scene-level factors relating to appearance. This can be seen from the fact that for datasets without ablations in appearance, *i.e.*, the (S+U) ablated datasets, the object segmentation performance is inferior. By contrast, the object segmentation accuracy can be greatly improved on the datasets only

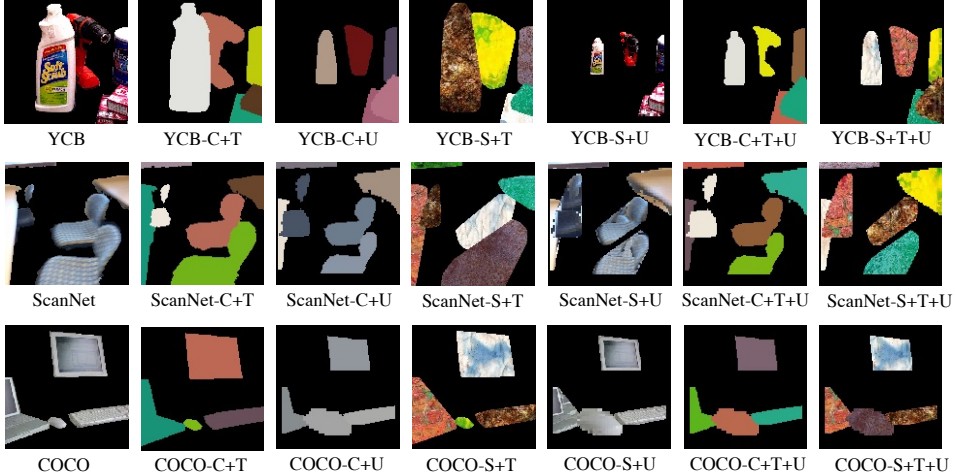

Figure 26: Example images of additional datasets ablated with both object- and scene-level factors.

Table 8: Quantitative results on additional datasets ablated with both object- and scene-level factors. Standard deviations of performance are calculated over 3 runs (marked with blue).

| | YCB-C+T | ScanNet-C+T | COCO-C+T |
|---|---|---|---|
| | AP / PQ / Pre / Rec | AP / PQ / Pre / Rec | AP / PQ / Pre / Rec |
| AIR [22] | 8.0 6.1 / 13.2 7.3 / 26.6 17.6 / 17.9 7.3 | 2.8 4.7 / 6.7 6.5 / 14.8 4.0 / 8.1 14.8 | 12.4 9.4 / 30.2 23.0 / 40.1 25.6 / 24.2 13.8 |
| MONet [8] | 82.9 9.6 / 66.7 0.5 / 73.0 1.2 / 87.8 12.2 | 36.5 28.4 / 33.5 21.8 / 40.4 24.1 / 50.3 19.7 | 66.1 5.6 / 53.2 3.7 / 56.2 2.3 / 74.4 6.4 |
| IODINE [25] | 78.9 2.6 / 59.7 2.1 / 71.5 0.6 / 83.6 2.7 | 65.1 2.9 / 51.1 1.0 / 62.4 2.9 / 74.2 0.5 | 55.1 10.4 / 42.2 7.5 / 56.3 6.4 / 66.2 9.2 |
| SlotAtt [40] | 58.7 12.1 / 43.2 5.0 / 57.8 9.4 / 71.0 9.4 | 29.2 2.9 / 27.6 1.3 / 34.4 1.4 / 49.2 1.2 | 22.1 8.6 / 19.6 4.1 / 25.2 4.1 / 35.8 8.1 |
| | YCB-C+U | ScanNet-C+U | COCO-C+U |
| | AP / PQ / Pre / Rec | AP / PQ / Pre / Rec | AP / PQ / Pre / Rec |
| AIR [22] | 11.4 4.9 / 16.4 5.4 / 25.8 8.7 / 25.1 6.5 | 5.4 0.5 / 12.3 2.6 / 35.1 15.3 / 12.6 0.4 | 20.5 10.8 / 24.9 7.1 / 47.7 10.2 / 30.5 8.8 |
| MONet [8] | 49.1 3.5 / 44.6 4.1 / 60.6 3.2 / 52.5 3.3 | 31.9 5.7 / 33.7 5.7 / 46.8 5.4 / 35.7 8.7 | 32.6 11.1 / 28.5 12.1 / 34.9 14.2 / 39.0 18.5 |
| IODINE [25] | 66.3 0.1 / 54.6 0.0 / 71.3 4.1 / 70.0 0.2 | 34.8 20.9 / 30.0 14.7 / 36.4 20.9 / 47.2 14.0 | 44.4 8.9 / 32.6 7.9 / 41.7 12.3 / 51.1 7.6 |
| SlotAtt [40] | 45.6 7.9 / 31.2 4.2 / 41.9 7.7 / 51.3 6.4 | 30.4 8.9 / 24.2 6.2 / 33.8 9.5 / 40.6 11.0 | 12.7 7.7 / 11.3 3.7 / 15.8 5.0 / 23.5 6.8 |
| | YCB-S+T | ScanNet-S+T | COCO-S+T |
| | AP / PQ / Pre / Rec | AP / PQ / Pre / Rec | AP / PQ / Pre / Rec |
| AIR [22] | 9.3 7.8 / 13.4 8.4 / 21.9 14.3 / 21.2 12.3 | 3.2 0.5 / 7.7 0.6 / 18.5 6.0 / 8.9 6.0 | 10.5 5.4 / 18.2 7.2 / 4.3 23.2 / 20.3 7.3 |
| MONet [8] | 86.7 2.2 / 78.9 1.6 / 81.9 1.8 / 87.0 2.2 | 83.2 3.7 / 77.8 13.0 / 86.0 20.0 / 83.8 2.9 | 80.2 0.0 / 68.5 7.2 / 73.2 11.7 / 82.3 1.0 |
| IODINE [25] | 41.9 17.9 / 33.2 11.4 / 41.5 12.0 / 51.3 14.7 | 54.9 23.1 / 44.4 16.2 / 52.0 16.7 / 68.0 21.6 | 44.1 1.1 / 34.3 0.7 / 41.0 0.3 / 56.8 0.6 |
| SlotAtt [40] | 77.3 7.8 / 60.6 1.5 / 75.0 2.3 / 69.2 19.7 | 24.9 46.1 / 21.2 35.0 / 25.5 38.3 / 37.9 43.2 | 67.4 3.1 / 50.2 1.6 / 59.2 1.6 / 76.4 0.3 |
| | YCB-S+U | ScanNet-S+U | COCO-S+U |
| | AP / PQ / Pre / Rec | AP / PQ / Pre / Rec | AP / PQ / Pre / Rec |
| AIR [22] | 0.8 5.4 / 2.9 7.1 / 5.0 10.0 / 5.2 12.7 | 6.3 0.2 / 13.1 3.3 / 32.4 16.3 / 14.3 1.2 | 20.0 9.5 / 25.8 10.0 / 48.3 15.6 / 31.6 10.9 |
| MONet [8] | 5.5 1.2 / 9.8 0.4 / 14.1 0.8 / 17.0 0.2 | 36.9 2.9 / 31.4 2.6 / 38.2 3.6 / 50.1 0.1 | 26.1 3.5 / 23.8 1.8 / 29.6 1.6 / 41.2 10.0 |
| IODINE [25] | 2.8 2.4 / 4.6 2.0 / 7.2 2.9 / 8.9 3.6 | 17.3 2.4 / 17.8 1.4 / 24.1 1.8 / 31.6 2.1 | 5.7 0.5 / 8.7 1.7 / 1.3 13.7 / 16.3 2.4 |
| SlotAtt [40] | 36.2 3.3 / 23.8 4.1 / 33.6 5.5 / 45.6 1.7 | 21.1 0.9 / 18.9 0.2 / 26.2 0.1 / 32.5 2.0 | 12.7 7.1 / 12.1 2.3 / 16.4 2.1 / 24.7 5.2 |
| | YCB-C+T+U | ScanNet-C+T+U | COCO-C+T+U |
| | AP / PQ / Pre / Rec | AP / PQ / Pre / Rec | AP / PQ / Pre / Rec |
| AIR [22] | 18.2 12.0 / 20.5 7.9 / 32.5 13.5 / 30.9 9.0 | 13.5 6.6 / 20.2 9.1 / 42.4 25.5 / 24.1 5.6 | 24.2 11.1 / 28.3 8.5 / 50.5 13.1 / 36.4 9.7 |
| MONet [8] | 63.2 6.9 / 66.5 11.8 / 86.3 13.0 / 64.5 5.7 | 74.6 5.6 / 62.1 6.6 / 67.8 7.2 / 79.2 3.2 | 73.0 3.3 / 75.7 10.3 / 94.2 23.1 / 73.1 7.5 |
| IODINE [25] | 54.9 23.1 / 38.7 18.2 / 58.0 12.0 / 59.4 20.8 | 65.5 3.8 / 48.0 3.2 / 59.8 5.9 / 70.7 2.9 | 47.9 11.7 / 35.5 7.4 / 53.8 2.1 / 53.4 14.2 |
| SlotAtt [40] | 73.8 4.6 / 53.0 2.9 / 67.6 7.8 / 75.2 4.5 | 58.5 8.1 / 45.5 10.2 / 52.9 8.9 / 68.3 11.9 | 30.5 27.6 / 20.3 18.0 / 27.8 23.0 / 38.9 23.3 |
| | YCB-S+T+U | ScanNet-S+T+U | COCO-S+T+U |
| | AP / PQ / Pre / Rec | AP / PQ / Pre / Rec | AP / PQ / Pre / Rec |
| AIR [22] | 24.6 20.0 / 24.8 13.7 / 37.4 20.7 / 37.6 18.6 | 16.7 10.1 / 23.9 12.0 / 47.8 30.2 / 28.5 7.8 | 29.4 16.5 / 34.6 18.4 / 58.3 35.7 / 42.2 13.4 |
| MONet [8] | 88.4 0.6 / 84.6 1.5 / 92.0 1.5 / 88.6 0.7 | 97.8 0.3 / 87.3 0.8 / 85.4 0.1 / 98.1 0.4 | 86.9 0.1 / 83.8 11.6 / 89.9 13.1 / 87.0 1.5 |
| IODINE [25] | 42.3 15.0 / 32.1 9.7 / 41.0 11.5 / 52.0 15.7 | 39.1 4.7 / 29.9 1.8 / 41.8 7.2 / 47.9 2.4 | 64.3 9.7 / 45.1 6.9 / 51.4 5.6 / 72.0 11.7 |
| SlotAtt [40] | 88.6 6.5 / 67.0 9.6 / 80.9 13.0 / 89.6 5.6 | 82.6 8.5 / 61.5 10.1 / 73.2 10.2 / 83.9 7.7 | 84.2 0.9 / 59.6 6.8 / 74.7 3.8 / 85.3 0.6 |

with appearance factors ablated, *i.e.*, the (C+T) datasets. Meanwhile, more regular shapes and uniform scales of objects still have a significant positive influence on the success of object segmentation especially when the appearance factors are combined in ablated datasets. To be specific, AIR [22] is quite sensitive to the scale of objects apart from object color gradient and inter-object color similarity. MONet [8] can obtain comparable performance to the simple synthetic datasets once object color gradient and inter-object color similarity are ablated. All four factors are closely relevant the results of IODINE [25] and SlotAtt [40].

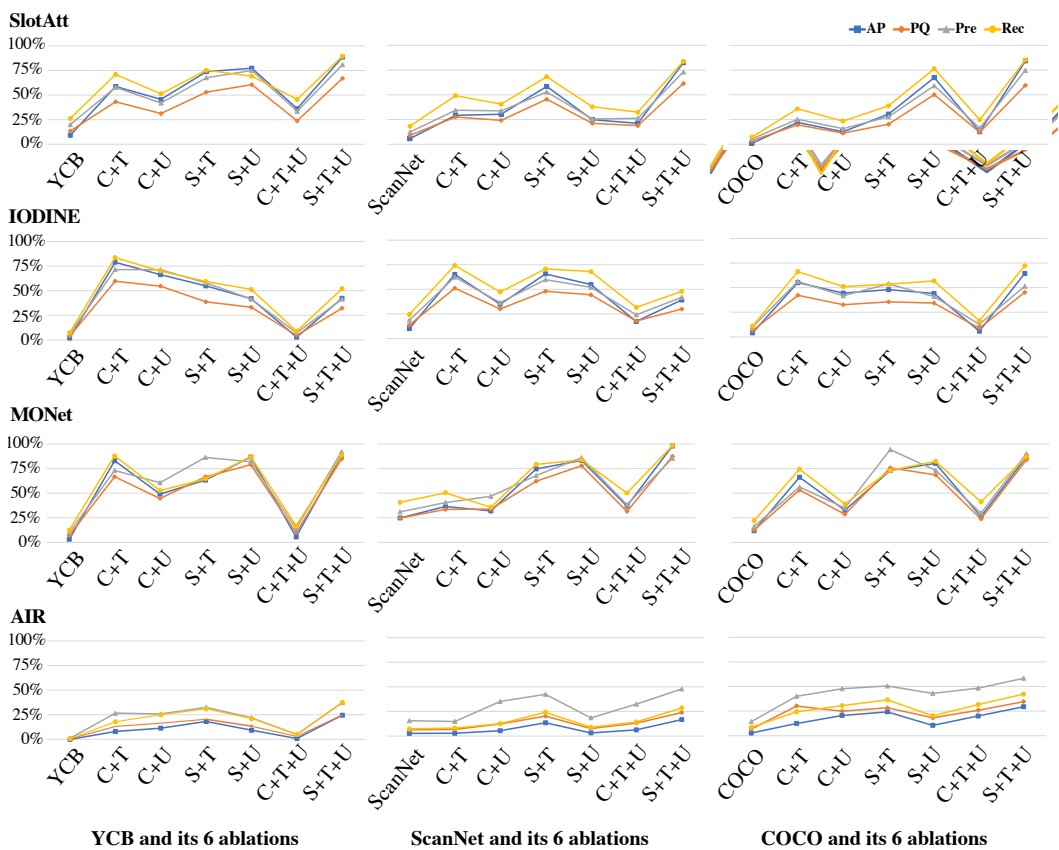

Figure 27: Quantitative results of baselines on three real-world datasets and their variants in Sec A.6.4

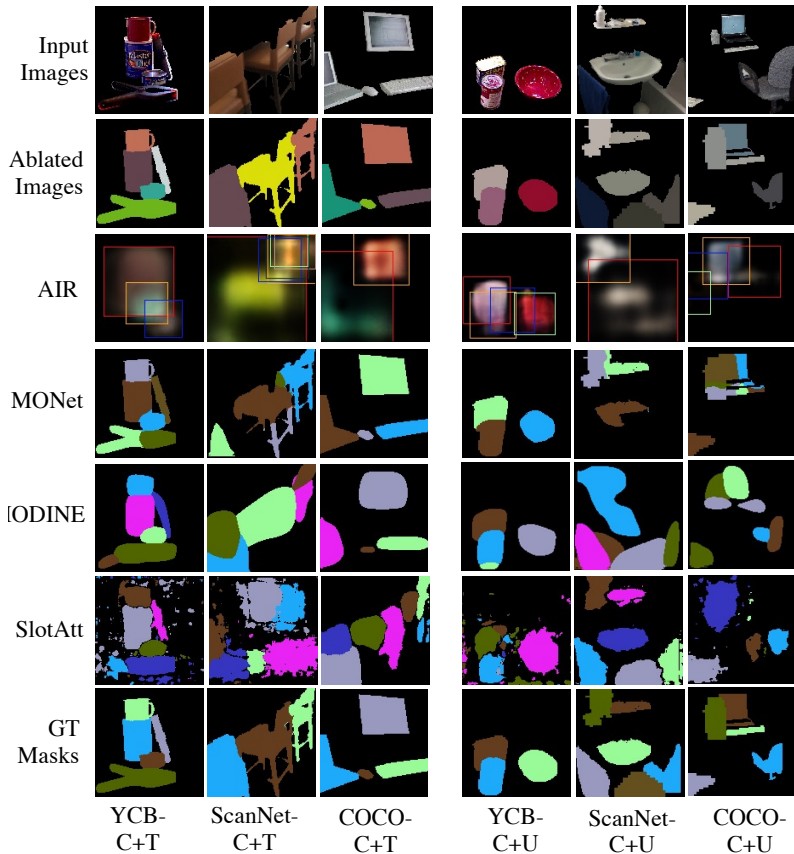

Figure 28: Qualitative results on the additional datasets ablated with object- and scene-level factors.

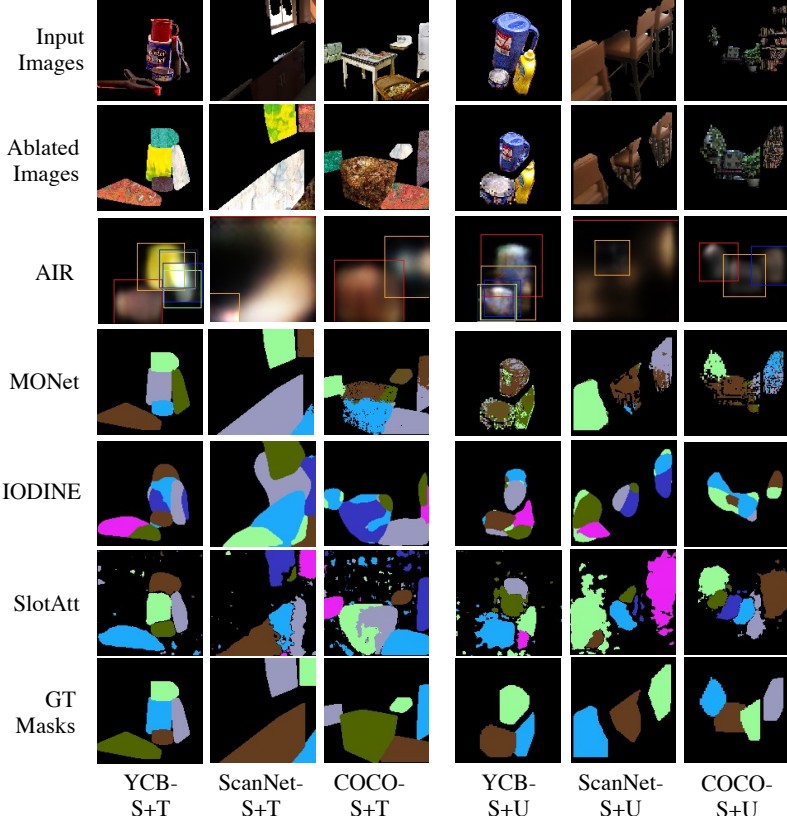

Figure 29: Qualitative results on the additional datasets ablated with object- and scene-level factors.

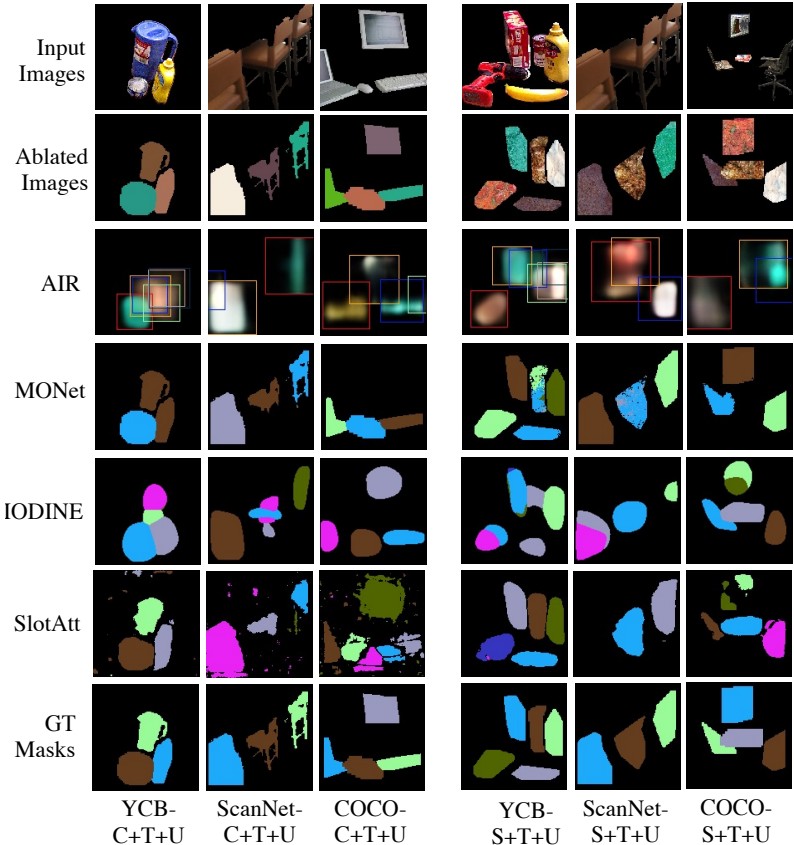

Figure 30: Qualitative results on the additional datasets ablated with object- and scene-level factors.