# OpenReview forum: "Promising or Elusive? Unsupervised Object Segmentation from Real-world Single Images"
_NeurIPS.cc/2022/Conference — NeurIPS 2022 Accept_

### Official Review · Reviewer_5Tue · 2022-07-11

**Rating:** 7
**Confidence:** 5
**Soundness:** 3 good
**Presentation:** 2 fair
**Contribution:** 3 good

**Summary:**

The paper presents an analysis of current unsupervised object-centric learning methods, focussing on shortcomings on real-world datasets. Since the performance of these methods on real images is very low, the paper changes the datasets using GT information, effectively simplifying the images. Analyzing the performance gain of specific alterations (e.g., uniform color, convex shape, etc.) give hints on which factors are especially difficult for current methods.

**Questions:**

The following questions are mainly derived from the weaknesses above.
1. How does the work position itself into related literature on benchmarking unsupervised object centric learning methods? Which findings are similar and which differ from previous work?

2. Are there findings about individual methods or approaches that generalize across the experiments? E.g., do VAE based methods generally perform better on texture-less scenes? Why?

3. Is there a way to present the results more clearly? For example by showing deltas in performance, in the same plot and without mixing bar charts and tables.

4. As all methods almost completely fail on original real-world data, are there any substantial findings that are directly related to real images? (Other than: it does not work)

5. What is the range of performance that can be expected across runs of a method? ClevrTex shows high std. dev. results for some methods, indicating that performance can strongly vary across runs.

6. Will the datasets, code, and models be published to help future methods improve and analyze their performance?



**Limitations:**

While the paper analyses the limitations of existing methods, the limitations of the presented approach are not discussed sufficiently. What insights cannot be gained from the proposed approach? What is the limit of the statements that can be derived with such a methodology?
Societal impact is low and adequatly discussed.

**Strengths And Weaknesses:**

Strengths
The paper shows how existing datasets can be altered to analyze the performance of current methods.

The experiments are methodical and the structure of the paper shows the findings in a logical order.

The changes in the datasets are motivated by intuitive Gestalt principles, resulting in findings that can potentially help the community move towards real-world application of these methods.

Weaknesses

Comparison and discussion of similar work:
Karazija L, Laina I, Rupprecht C. ClevrTex: A Texture-Rich Benchmark for Unsupervised Multi-Object Segmentation, NeurIPS Datasets&Benchmarks 2021.
Proposes a benchmark and an analysis of a similar set of methods. The motivation for introducing a new dataset is based on the same intuition: current real-world datasets are too difficult. Additionally it comes to some of the same conclusions (e.g., texture vs. lighting vs. plain colors).
Weis MA, Chitta K, Sharma Y, Brendel W, Bethge M, Geiger A, Ecker AS. Benchmarking Unsupervised Object Representations for Video Sequences. J. Mach. Learn. Res.. 2021
Analysis of similar concepts for video segmentation methods with a focus on occlusions. Many of the investigated methods are video extensions of the approaches in this paper.

Discussion of the differences between methods: It could be interesting to also analyze the differences between methods. For example AIR does not seem to be affected by any of the changes and does not perform well, whereas MONet  benefits from almost all changes (except YCB-S+C+U – why?) Since methods differ substantially, specific insights can be very valuable to understand what to improve.

Clarity of presentation: The results could be presented in a more comprehensive way. I found myself often confused about what to compare to what. Examples:
- Color ablation datasets have the suffix S, whereas shape ablation has the suffix C.
- To understand the changes in Figure 4 one has to compare the second half of each plot to the second half of the plots in Fig 3. Additionally, the order in Fig 3 and 4 is different.
- To understand the results in Fig 5,7,8 (which are bar plots from 0-1) one has to look up the numerical values in Tab.3 (which are numbers from 0-100).
This makes it often very difficult to follow the reasoning in the paper and obfuscates the message.

---

> ### Author Response · Authors · 2022-08-02
> **Responses to Reviewer 5Tue (Part 1)**
>
> We appreciate the reviewer's insightful comments and address the main concerns below.
>
> **Q1: Comparison and discussion of similar work ... of the approaches in this paper.**
> Thanks for pointing out the recent relevant papers. As suggested, in the revised paper, we have added a separate paragraph "Related Work" in Section 1 page 2.
>
> _CLEVRTEX, Karazija et.al.,_: As a benchmark for unsupervised object segmentation, it shares similarities with our work. Both conduct extensive experiments on the-state-of-art models on a set of benchmark datasets. However, CLEVRTEX focuses on the characteristics and comparison of different models. Our work, on the other hand, aims to quantify properties/inductive biases of synthetic and real-world datasets, and then discover what dataset factors incur the failure of existing models on challenging images. Notably, we employ real-world datasets instead of only complex synthetic datasets for systematically evaluation.
>
> _Benchmarking Unsupervised.., Weis et.al.,_: This work targets at object-centric learning on sequential data instead of single images. It shows detailed architectural analysis and comparison between a set of models. Similar to our work, they also consider properties of datasets. Specifically, several challenging cases such as occlusions are discussed. However, they are still limited to synthetic datasets, and its dataset properties are generally descriptive rather than quantitative.
>
> **Q2: Discussion of the differences between methods ... what to improve.**
>
> This is a great question. We have added an additional section (Section 4.5) in the revised main paper to analyze the sensitivity of different models on different dataset factors. In particular,
> - **AIR**:  As a factor based model, AIR has a strong spatial-locality bias. Despite its poor segmentation performance across all datasets, there is a notable improvement when inter-object shape variation is ablated from real-world datasets (U / T+U / S+C+U / S+C+T+U). More convincingly, even when all other three factors are ablated (S+C+T), it can be hardly improved. These observations show that object shape variation is a significant factor for AIR.
> - **MONet**:  MONet is more sensitive to color-related factors than shape-related factors. The ablations of object color gradient and inter-object color similarity significantly improve its performance, while ablations of object shape concavity and inter-object shape variation make little differences. For the two color-related factors, the scene-level one is more important than the object-level factor. From this, we can see that MONet has a strong dependency on color. Similar colors tend to be grouped together while different colors are separated apart. Furthermore, the ablation on object color gradient alleviates over-segmentation whereas the ablation on inter-object color similarity alleviates under-segmentation. We conjecture that under-segmentation can be a more severe issue for MONet on real-world datasets, leading to a larger sensitivity on the scene-level color factor.
> - **IODINE**: IODINE also has a heavy dependency on both object- and scene-level color-related factors. However, different from MONet, the ablation on object color gradient brings better performance than inter-object color similarity. We speculate it is because the regularization on shape latent alleviates under-segmentation by bias towards more regular shapes. In this way, over-segmentation is the key issue, making the inter-object color similarity a dominant factor.
> - **SlotAtt**: The ablations on all four factors increase the performance of SlotAtt at different levels, among which object- and scene-level color-related factors are more significant. We conjecture that it is because the feature embeddings used by Slot Attention module are learnt from both pixel colors and coordinates, which contributes to its sensitivity to both shape and color factors.
>
> Overall, we hope that these findings could be beneficial for future studies in the community.
>
>
>
> **Q3: Clarity of presentation: ..**
>
> **Q3.1: Color ablation datasets have the suffix S, whereas shape ablation has the suffix C.**
>
> Thanks for the advice and we have inter-changed S and C for better understanding.
> The ablation abbreviation look-up table is summarized here. This will be added into appendix:
>
> | ablation  | object-level | scene-level | color-related | shape-related | Object Color Gradient | Object Shape Concavity | Inter-object Color Similarity | Inter-object Shape Variation |
> | ------------- | ------------- | ------------- | ------------- | ------------- | ------------- | ------------- | ------------- | ------------- |
> | C  | &check; |  | &check; |  | &check; | | | |
> | S  | &check; |  |  | &check; |  | &check; | | |
> | T  |  | &check; | &check; |  |  | |&check; | |
> | U |  | &check; |  | &check; |  | | | &check;|

---

> > ### Author Response · Authors · 2022-08-02
> > **Responses to Reviewer 5Tue (Part 2)**
> >
> > continued with Part 1
> >
> > **Q3.2: To understand the changes in Figure 4 ... is different.**
> >
> > All complexity factor distributions are now shown in a single figure (Figure 5 in the revised paper). The first row presents distributions of complexity factors on three synthetic and original real-world datasets. The second row presents distributions of complexity factors on three synthetic and real-world datasets ablated on object-level factors. The third presents distributions of complexity factors on three synthetic and real-world datasets ablated on scene-level factors.
> >
> > **Q3.3: To understand the results in Fig 5,7,8 ... obfuscates the message.**
> >
> > Thanks for the nice suggestion. In Figure 4 of the revised submission, quantitative results of five methods on six datasets are presented with a bar chart. In Figure 6, we have grouped all ablation experiments. It contains results of four methods on three types of real-world datasets. Each sub-figure contains one original and nine ablated versions of a dataset.
> >
> > **Q4: How does the work position itself  ... from previous work?**
> >
> > In summary, the existing relevant works on benchmarking unsupervised object-centric learning focus on characterizing and analyzing architectural design of different models, and their experiments are still limited to synthetic datasets. By comparison, our work targets at real-world datasets. Since all mentioned models fail on real-world datasets, architectural analysis is barely enough. Instead, we summarize and quantify inductive biases across different datasets. From our experiments, we find that different models present different sensitivity to different dataset properties/biases, which also validates the findings of other study. More importantly, with the study of objectness biases in datasets, it is expected that better formulations of object-centric learning can be inspired in the future especially in the context of real-world applications.
> >
> > **Q5: Are there findings about individual methods ... Why?**
> >
> > We have made the following observations based on current experimental results:
> > - Factor-based models exhibit a higher sensitivity to scene-level shape factor than layer-based models. AIR, as a factor-based model, can obtain a better segmentation performance when the inter-object shape variation is ablated from real-world datasets. The result of other three layer-based models, however, are less relevant to the scene-level shape factor. We believe it is because objects are  represented by explicit factors (such as scale, position, appearance) in a factor-based model, and the scene is then modelled as a spatial combination of objects. Such design forces objects to be bounded within a region. In contrast, in layer-based models where objects are modeled by masks, there are less spatial-locality constraints.
> > - Layer-based models are more more sensitive to object color gradient and inter-object color similarity. As discussed above, layer-based models have a less constraint on spatial-locality of objects, which allows more flexible pixel clustering. Thus, the segmentation decision has a higher dependency on color information. If an object is too colorful, it is likely to be segmented into multiple components. If objects in a scene are too similar in terms of color, they can be hard to be segmented apart.
> >
> > **Q6: Is there a way ... tables.**
> >
> > Well-addressed above. Refer to **Q3** above.
> >
> > **Q7: As all methods almost completely  ... does not work)**
> >
> > This is great question. In addition to our analysis and findings for each of the four models in the newly added Section 4.5, we further conduct the following generalization experiments to investigate how the real images impact the models. In particular, we use the well-trained model from dSprites dataset to directly test on three fully-ablated real-world datasets, i.e., removing all four factors. The quantitative results are as follows:
> >
> >
> > | dataset  | AIR | MONet | IODINE | SlotAtt |
> > | ------------- | ------------- | ------------- | ------------- | ------------- |
> > |   | AP / PQ / Pre / Rec | AP / PQ / Pre / Rec | AP / PQ / Pre / Rec | AP / PQ / Pre / Rec |
> > | YCB - C+S+T+U | 21.0 / 25.4 / 42.2 / 37.1 | 69.5 / 56.4 / 64.1 / 77.0 | 87.2 / 65.5 / 80.6 / 89.7 | 67.5 / 50.3 / 58.5 / 75.4 |
> > | ScanNet - C+S+T+U  | 12.8 / 14.7 / 27.3 / 20.5 | 54.8 / 45.3 / 50.9 / 68.9 | 65.7 / 49.0 / 63.5 / 71.8 | 23.4 / 24.5 / 34.1 / 39.0 |
> > | COCO - C+S+T+U  | 18.0 / 21.5 / 40.7 / 28.1 | 71.3 / 54.5 / 57.5 / 79.1 | 71.7 / 56.7 / 71.7 / 81.5 | 52.0 / 41.9 / 51.8 / 61.2 |
> >
> > We can see that the model trained on dSprites dataset achieves excellent segmentation results on the three fully-ablated real-world datasets. It means that the ablated real-world datasets indeed share similar distributions with synthetic dataset dSprites. This again confirms that the complex distributions of object- and scene-level color and shape factors in real-world images are particularly diverse and challenging, incurring the failure of existing models.

---

> > > ### Author Response · Authors · 2022-08-02
> > > **Responses to Reviewer 5Tue (Part 3)**
> > >
> > > continued with Part 2
> > >
> > > **Q8:  What is the range of performance ... across runs.**
> > >
> > > As suggested, we run four unsupervised models on the three synthetic datasets and three original real-world datasets for three times. The table below shows the results and ***standard deviations***.  You may refer to Table 3 in revised appendix for better visualization.
> > >
> > > | dataset  | AIR | MONet | IODINE | SlotAtt |
> > > | ------------- | ------------- | ------------- | ------------- | ------------- |
> > > |   | AP / PQ / Pre / Rec | AP / PQ / Pre / Rec | AP / PQ / Pre / Rec | AP / PQ / Pre / Rec |
> > > | dSprites | 45.4 ***1.8*** / 38.2 ***3.0*** / 57.6 ***7.4*** / 58.1***7.5*** | 69.7 ***4.1*** / 61.6 ***6.0*** / 70.4 ***8.1*** / 73.9 ***1.9***  | 92.9 ***4.3*** / 71.3 ***6.1*** / 82.6 ***2.3*** / 96.0 ***5.2*** | 92.9 ***1.4*** / 82.8 ***1.6*** / 88.8 ***3.4*** / 92.9 ***1.6*** |
> > > | Tetris  | 25.2 ***13.9*** / 23.4 ***12.4*** / 36.8 ***20.9*** / 39.9 ***12.9*** | 85.9 ***13.0*** / 75.8 ***13.6*** / 85.1 ***16.4*** / 89.7 ***8.2***| 52.2 ***2.3*** / 37.9 ***4.6*** / 48.0 ***2.3*** / 61.8***1.7*** |  94.3 ***1.2*** / 79.9 ***6.4*** / 90.5 ***3.3*** / 94.4 ***1.3***|
> > > | CLEVR  |  46.4 ***14.0*** / 44.3 ***12.4*** / 67.4 ***9.9*** / 52.5 ***15.9*** | 39.0 ***8.5*** / 37.3 ***6.3*** / 65.6 ***11.8*** / 42.8 ***10.8***| 82.8 ***2.8*** / 73.0 ***5.7*** / 77.5 ***3.1*** / 87.4 ***2.0*** |  91.7 ***6.4*** / 82.9 ***10.9*** / 90.8 ***9.7*** / 92.7 ***5.3***|
> > > | YCB |   0.0 ***0.1*** / 0.6 ***0.3*** / 1.1 ***0.4*** / 0.8 ***0.2*** | 3.1 ***1.6*** / 7.0 ***2.6*** / 9.8 ***3.6*** / 1.2 ***0.8***| 1.8 ***0.2*** / 3.9  ***1.3*** / 6.2 ***2.0*** / 7.3 ***1.9*** |  9.2 ***0.4*** / 13.5 ***0.9*** / 20.0 ***1.3*** / 26.2 ***6.8***|
> > > | ScanNet  | 2.7 ***1.4*** / 6.3 ***1.7*** / 15.6 ***2.8*** / 7.3 ***1.6*** | 24.8 ***1.6*** / 24.6***1.6*** / 31.0 ***1.6*** / 40.7 ***1.8***| 10.1 ***2.9*** / 13.7 ***2.7*** / 18.6 ***4.2*** / 24.4 ***3.8*** |  5.7 ***0.3*** / 9.0 ***1.5*** / 12.4 ***2.5*** / 18.3 ***2.7***|
> > > | COCO  |   2.7 ***0.1*** / 6.7***0.5*** / 14.3 ***2.6*** / 8.6 ***0.8*** | 11.8 ***2.0*** / 12.5 ***1.1*** / 16.1 ***0.9*** / 21.9 ***1.7***| 4.0 ***1.2*** / 6.3 ***1.2*** / 9.9 ***1.8*** / 10.8 ***2.0*** |  0.8 ***0.3*** / 3.5 ***1.2*** / 5.3 ***1.7*** / 7.3 ***2.2***|
> > >
> > >
> > > It is observed that AIR and MONet have a larger variance of performance. We conjecture that this is because AIR and MONet have more hyperparameters to select such as the weights for regularization and prior distribution. In addition, real-world datasets generally have smaller standard deviations, primarily because all models fail on the real-world datasets and such standard deviations do not have valuable meanings.
> > >
> > > **Q9: Will the datasets, code, and models be published to help future methods improve and analyze their performance?**
> > >
> > > Yes.

---

> > > > ### Comment · Reviewer_5Tue · 2022-08-08
> > > > **Thanks for the replies!**
> > > >
> > > > Thank you for the comprehensive replies and extensive additional evaluation. The changes to the figures and tables improve the readability and the additional experiments are convincing.
> > > > The analysis of the differences between models is quite interesting and helps the message of the paper.
> > > > I did not find any unadressed major concerns in the other reviews and am happy to recommend acceptance.

---

> > > > > ### Author Response · Authors · 2022-08-08
> > > > > **Thanks**
> > > > >
> > > > > We really appreciate your encouraging feedback.

---

### Official Review · Reviewer_Zitt · 2022-07-11

**Rating:** 7
**Confidence:** 4
**Soundness:** 3 good
**Presentation:** 2 fair
**Contribution:** 3 good

**Summary:**

Many object-centric methods have been proposed during the last years that learn to
segment scenes and represent the objects individually. These models however don't scale
to realistic data yet. In the present study, the authors seek to understand which
factors contributing to the complexity of real world images have a high influence on
model performance. To that end, the authors train 4 object-centric models on different
variants of several datasets in which the complexity factors are manipulated
individually. The authors find that both object-level and scene-level factors contribute
to the difficulties that recent models have on natural data and conclude that essential
inductive biases are missing in recent models.

**Questions:**

- How does the present study compare to other studies anlysing object-centric models? To
  what degree do your results confirm or reject previous results? The paper should
  contain a related work section adressing these questions. Other studies that should
  be mentioned are, e.g., [Karazija et al. 2021](http://arxiv.org/abs/2111.10265) and
  [Papa et al. 2022](http://arxiv.org/abs/2204.08479).

**Limitations:**

Limitations are not explicitely discussed in the paper, and the paper would be
strengthened by extending the discussion in that respect.

**Strengths And Weaknesses:**

Many papers exist that propose new object-centric models, but only few that analyse the
strenghts and weaknesses of the existing methods in greater detail. The problem that
recent models don't scale to realistic data is well known but not solved yet. Studies
that test models in settings with controlled complexity to better understand these
shortcomings are therefore of great interest for the field. A particular strength of the
paper is the very systematic way of varying the several complexity factors for creating
the dataset veriants.

A weakness of the paper in my view is the presentation of the setup and the results.
Naturally, a study as the one presented in this paper has a certain complexity and
produces many subresults. Presenting those in a way that makes it easier to grasp the
major results would support the paper by making it more accessible. For example:
- For each factor, showing samples with different complexity values would make it
  easier to understand the individual factors and the complexity ranges oberved in
  the datasets used.
- The results of the different variants of a single dataset are divided over many
  figures. Comparing the performances of each model for the differently ablated datasets
  is a key endeavor of the paper, the respective results are however not presented
  jointly. Grouping the results differently might improve the paper in this respect,
  e.g. by having a figure with performances as bar plots, that has models on the x-axis
  and all variants of a single dataset on the y-axis.

---

> ### Author Response · Authors · 2022-08-02
> **Responses to Reviewer Zitt**
>
> We appreciate the reviewer's insightful comments and address the main concerns below.
>
> **Q1: For each factor, showing samples with different complexity values would make it easier to understand the individual factors and the complexity ranges observed in the datasets used.**
>
> In the revised paper, we have added Figure 3 in page 4 to clearly show sample images for the four complexity factors at different values.
>
> **Q2: The results of the different variants ...  of a single dataset on the y-axis.**
>
> Thanks for the great point. In the revised paper, we carefully combine all ablation results in a single figure (Figure 6 on page 7). Indeed, the results are way better to compare and analyze than the previous multiple plots.
>
> **Q3: How does the present study compare to e.g., Karazija et al. 2021 and Papa et al. 2022.**
>
> Thanks for pointing out the recent relevant papers. As suggested, in the revised paper, we have added a separate paragraph "Related Work" in Section 1 page 2.
>
> _CLEVRTEX, Karazija et.al.,_: As a benchmark for unsupervised object segmentation, it shares similarities with our work. Both conduct extensive experiments on the-state-of-art models on a set of benchmark datasets. However, CLEVRTEX focuses on the characteristics and comparison of different models. Our work, on the other hand, aims to quantify properties/inductive biases of synthetic and real-world datasets, and then discover what dataset factors incur the failure of existing models on challenging images. Notably, we employ real-world datasets instead of only complex synthetic datasets for systematically evaluation.
>
> _Inductive Biases.., Papa et.al.,_: This paper presents a very detailed study on the performance of MONet and SlotAtt on several synthetic datasets, so as to analyze architectural biases in the design of both models. Our work has similar findings on the inductive biases of MONet and SlotAtt, which can be found in the newly added Section 4.5. Apart from the study on biases of SOTA models with synthetic datasets, our work also analyzes their failure on real-world datasets with extensive ablation experiments.
>
> In summary, in the two relevant works, their experiments are still limited to synthetic images and they tend to characterize and analyze architectural designs of different models. By comparison, our work benchmarks existing models on real-world datasets. Since all mentioned models fail on real-world datasets, architectural analysis is barely enough. Instead, we summarize and quantify inductive biases across different datasets. From our experiments, we find that different models present different sensitivity to different dataset properties/biases, which also validates the findings of other study. More importantly, with the study of objectness biases in datasets, it is expected that better formulations of object-centric learning can be inspired in the future especially in the context of real-world applications.

---

> > ### Comment · Reviewer_Zitt · 2022-08-10
> > **Re: Responses to Reviewer Zitt**
> >
> > Thank you for your detailed response and the revision.
> >
> > In my view the clarity of the paper has been clearly improved. The new Figure 6 and the model-wise discussion make it easier to get a general impression of the influence of different factors on the model performances. Overall I think there is still room for improving the clarity: It find it somewhat challenging to keep track of all necessary details when reading the paper and hard to get the gist of the work when only briefly skimming the paper.
> >
> > However, content-wise I am still convinced that this study is valuable for the field. And the clarity has been improved with the revision, so I am happy to increase my rating of the paper.

---

### Official Review · Reviewer_fjR6 · 2022-07-15

**Rating:** 6
**Confidence:** 5
**Soundness:** 4 excellent
**Presentation:** 4 excellent
**Contribution:** 3 good

**Summary:**

This paper keys in on something that is widely suspected, but has not been rigorously investigated in the unsupervised object segmentation / discovery literature: the fact that nearly all methods do well on "simple" synthetic datasets but not on complex, real-world datasets that are the subject of most supervised object segmentation work.

In particular, this work makes two contributions:

1. It tests several of the better and more popular unsupervised object discovery methods on a range of datasets, spanning simple/synthetic to complex/real-world, and (maybe for the first time) actually quantifies the performance on the harder datasets using standard metrics. (For some reason, a lot of the unsupervised object discovery literature uses the non-standard foreground-ARI metric, which the authors here correctly note will make most models look far better quantitatively than they are qualitatively.)

In this experiment, the authors find -- unsurprisingly -- that the models they test fail catastrophically on real-world data.

2. The paper creates new datasets in which the real-world datasets are "ablated" to alter object- and scene-level factors, which the authors hypothesize could explain the difference in model performance on synthetic vs real-world data. The authors then test this by training/testing the models on the new "ablated" datasets. When some or all of the factors that distinguish the real-world data are ablated, the unsupervised models can actually perform quite well.

**Questions:**

Everything in the paper was clear. If the authors could address the weaknesses above, I think this would be a valuable contribution to the literature even though it does not introduce a new algorithm.

**Limitations:**

Yes

**Strengths And Weaknesses:**

Strengths:

1. Actually quantifying the unsupervised model performance on harder data was necessary, even if the results are unsurprising. This is an important contribution to the literature as it makes explicit something that many people "knew about," but could not point to concretely.

2. The dataset ablation experiments are very creative and the results are interesting. I was surprised that relatively simple factors could explain the gap in performance between synthetic and real datasets, although looking at the "ablated" images in Figure 9 it makes more sense: the fully-ablated datasets really do look much more like the synthetic datasets in terms of object and scene complexity and distribution.

3. The ideas and experiments are well-motivated and clearly explained.

Weaknesses:

1. It's hard to draw conclusions about what the *models* are missing because most of them have very similar design principles. The authors don't comment much on which models seem to handle which dataset factors better than others.

2. The authors suggest that models based on different principles/inductive biases might be needed to handle the complexity of real-world data, especially ones based on learning from object motion. There is some recent work in this direction (e.g. https://arxiv.org/abs/2205.08515 and https://arxiv.org/abs/2110.06562 and https://proceedings.neurips.cc/paper/2020/file/4324e8d0d37b110ee1a4f1633ac52df5-Paper.pdf). This may have been too recent for the authors to know about at the time of writing (and to test) but it's important to point out that there are ideas for dealing with these issues and some progress.

3. There is no comparison to supervised methods with the ablated (or original) datasets. This is important because it's unclear whether supervised methods perform better on real data due only to their receiving supervision or also because they have better architectures for the segmentation task. (Some standard architecture, like Mask-RCNN, could be useful here.)

4. Conversely, an important control would be to somehow provide supervision to one or more of the object discovery models. This would again help to dissociate the effects of their having a worse architecture from the effects of their using weak learning principles.

---

> ### Author Response · Authors · 2022-08-02
> **Responses to Reviewer fjR6**
>
> We appreciate the reviewer's insightful comments and address the main concerns below.
>
> **Q1: It's hard to draw conclusions about ... than others.**
>
> This is a great question. We have added an additional section (Section 4.5) in the revised main paper to analyze the sensitivity of different models on different dataset factors. In particular,
> - **AIR**:  As a factor based model, AIR has a strong spatial-locality bias. Despite its poor segmentation performance across all datasets, there is a notable improvement when inter-object shape variation is ablated from real-world datasets (U / T+U / S+C+U / S+C+T+U). More convincingly, even when all other three factors are ablated (S+C+T), it can be hardly improved. These observations show that object shape variation is a significant factor for AIR.
> - **MONet**:  MONet is more sensitive to color-related factors than shape-related factors. The ablations of object color gradient and inter-object color similarity significantly improve its performance, while ablations of object shape concavity and inter-object shape variation make little differences. For the two color-related factors, the scene-level one is more important than the object-level factor. From this, we can see that MONet has a strong dependency on color. Similar colors tend to be grouped together while different colors are separated apart. Furthermore, the ablation on object color gradient alleviates over-segmentation whereas the ablation on inter-object color similarity alleviates under-segmentation. We conjecture that under-segmentation can be a more severe issue for MONet on real-world datasets, leading to a larger sensitivity on the scene-level color factor.
> - **IODINE**: IODINE also has a heavy dependency on both object- and scene-level color-related factors. However, different from MONet, the ablation on object color gradient brings better performance than inter-object color similarity. We speculate it is because the regularization on shape latent alleviates under-segmentation by bias towards more regular shapes. In this way, over-segmentation is the key issue, making the inter-object color similarity a dominant factor.
> - **SlotAtt**: The ablations on all four factors increase the performance of SlotAtt at different levels, among which object- and scene-level color-related factors are more significant. We conjecture that it is because the feature embeddings used by Slot Attention module are learnt from both pixel colors and coordinates, which contributes to its sensitivity to both shape and color factors.
>
> Since existing models usually consist of multiple network components, to fully investigate the effectiveness of each individual building block involves comprehensive ablations on network components. This is left for the future work, and our paper focuses on extensive ablations and analysis from the perspective of dataset factors.
>
> **Q2: The authors suggest that models based on different principles/inductive biases ... and some progress.**
>
> Thanks for pointing out the latest and relevant research papers. In the revised paper (Section 5), we have explicitly discussed two potential future directions including the motion based pipeline.
>
> **Q3: There is no comparison to supervised methods ... like Mask-RCNN, could be useful here.)**
>
> Thanks for the advice. For a more comprehensive comparison, we include Mask-RCNN as an additional baseline for the main experiments on six datasets in Section 4.1. The quantitative results are as follows, and qualitative results are in appendix. As expected, we can see that the fully-supervised Mask-RCNN achieves nearly perfect segmentation scores on all three synthetic datasets, and very satisfactory results on the challenging real-world datasets. These scores can indeed help the audience to understand the current status of existing unsupervised models. Due to the time limitation, MaskRCNN results for all other ablation experiments are still cooking up.
>
> | dataset  | AP | PQ | Pre | Rec |
> | ------------- | ------------- | ------------- | ------------- | ------------- |
> | dSprites  | 98.4 | 90.2 | 99.6 | 98.4 |
> | Tetris  | 99.8 | 90.3 | 99.8 | 99.8 |
> | CLEVR  | 98.2 | 90.0 | 97.8 | 99.5 |
> | YCB  | 62.9 | 58.4 | 83.3 | 66.9 |
> | ScanNet  | 41.4 | 43.3 | 65.2 | 50.5 |
> | COCO  | 46.0 | 47.9 | 71.7 | 53.2 |
>
> **Q4: Conversely, an important control would be ... using weak learning principles.**
>
> Providing supervision to existing unsupervised methods is an interesting idea to validate the significance of supervision signals. Nevertheless, it is non-trivial to systematically modify the complex neural architecture of existing models and tune parameters.
>
> As also suggested by the reviewer, we believe that adding the fully-supervised Mask-RCNN as the additional baseline can be sufficient to demonstrate the performance gaps.

---

### Author Response · Authors · 2022-08-03
**Overall Response**

We appreciate insightful comments and valuable suggestions from all reviewers. A revised paper together with supplementary materials are presented. Specifically, we highlight the revised content with yellow color. Revisions include:

+ Presentation
	+ Sample images for the four complexity factors at different values (Figure 3).
	+ Change quantitative results table for original six datasets into a bar chart (Figure 4).
    + Group complexity factors for all ablation datasets into one figure (Figure 5).
    + Group quantitative results for all ablation datasets into one figure (Figure 6).
    + Change abbreviation names for ablation datasets (Table 4, Appendix).
+ Experiments
	+ Fully-supervised Mask R-CNN as an additional baseline (Figure 4; Figure 15, Appendix).
	+ Range of performance over 3 runs. (Table 3, Appendix)
+ Discussions
	+ Related work (Section 1, Page 2).
	+ Comparison of different models on their sensitivity to different dataset factors (Section 4.5).

---

### Meta-Review · Area_Chair_V9LW · 2022-08-27

**Recommendation:** Accept
**Confidence:** Certain

**Metareview:**

This paper conducts a systematic evaluation of existing unsupervised object segmentation methods on real-world images. Using ablated real-world datasets, the authors identify factors causing the failures of existing methods on real images.

All three reviewers find the study valuable and creative. This type of studies should be encouraged.

To address Reviewer 5Tue's concerns, the authors added an additional section (Section 4.5) to analyze the sensitivity of different models on different dataset factors. They also added comparison results. Reviewer Zitt felt that the clarity of the paper has been clearly improved. This reviewer was convinced that this study is valuable for the field. Reviewer 5Tue felt that the changes to the figures and tables improve the readability and the additional experiments are convincing. This reviewer did not find any unadressed major concerns.



**Award:**

No

---

### Decision · Program_Chairs · 2022-09-14

Accept